# Bayesian machine learning analysis of single-molecule fluorescence colocalization images

Yerdos A Ordabayev, Larry J Friedman, Jeff Gelles*, Douglas L Theobald*

Department of Biochemistry, Brandeis University, Waltham, United States

**Abstract** Multi-wavelength single-molecule fluorescence colocalization (CoSMoS) methods allow elucidation of complex biochemical reaction mechanisms. However, analysis of CoSMoS data is intrinsically challenging because of low image signal-to-noise ratios, non-specific surface binding of the fluorescent molecules, and analysis methods that require subjective inputs to achieve accurate results. Here, we use Bayesian probabilistic programming to implement Tapqir, an unsupervised machine learning method that incorporates a holistic, physics-based causal model of CoSMoS data. This method accounts for uncertainties in image analysis due to photon and camera noise, optical non-uniformities, non-specific binding, and spot detection. Rather than merely producing a binary 'spot/no spot' classification of unspecified reliability, Tapqir objectively assigns spot classification probabilities that allow accurate downstream analysis of molecular dynamics, thermodynamics, and kinetics. We both quantitatively validate Tapqir performance against simulated CoSMoS image data with known properties and also demonstrate that it implements fully objective, automated analysis of experiment-derived data sets with a wide range of signal, noise, and non-specific binding characteristics.

## Editor's evaluation

Using a Bayesian machine learning approach, the authors of this paper have developed a tool for the analysis of single-molecule fluorescence colocalization microscopy images. The authors develop the algorithm, generate an associated software program, and then benchmark the algorithm and software using both simulated and experimental data. The results provide an important, validated tool for use by the single-molecule fluorescence microscopy community.

*For correspondence:
gelles@brandeis.edu (JG);
dtheobald@brandeis.edu (DLT)

**Competing interest:** The authors declare that no competing interests exist.

## Introduction

A central concern of modern biology is understanding at the molecular level the chemical and physical mechanisms by which protein and nucleic acid macromolecules perform essential cellular functions. The operation of many such macromolecules requires that they work not as isolated molecules in solution but as components of dynamic molecular complexes that self-assemble and change structure and composition as they function. For more than two decades, scientists have successfully explored the molecular mechanisms of many such complex and dynamic systems using multi-wavelength single molecule fluorescence methods such as smFRET (single-molecule fluorescence resonance energy transfer) (*Roy et al., 2008*) and multi-wavelength single-molecule colocalization methods (CoSMoS, colocalization single molecule spectroscopy) (*Larson et al., 2014*; *van Oijen, 2011*; *Friedman and Gelles, 2012*).

CoSMoS is a technique to measure the kinetics of dynamic interactions between individual molecules. The CoSMoS method has been used for elucidating the mechanisms of complex biochemical

**Figure 1.** Example CoSMoS experiment. (**A**) Experiment schematic. DNA target molecules labeled with a blue-excited fluorescent dye (blue star) are tethered to the microscope slide surface. RNA polymerase II (Pol II) binder molecules labeled with a green-excited dye (green star) are present in solution. (**B**) Data collection and preprocessing. After collecting a single image with blue excitation to identify the locations of the DNA molecules, a time sequence of Pol II images was collected with green excitation. Preprocessing of the images includes mapping of the corresponding points in target and binder channels, drift correction, and identification of two sets of areas of interest (AOIs). One set corresponds to locations of target molecules (e.g., purple square); the other corresponds to locations where no target is present (e.g., yellow square). (**C**) On-target data. Data are time sequences of 14 × 14 pixel AOI images centered at each target molecule. Frames show presence of on-target (e.g., frame 630) and off-target (e.g., frame 645) Pol II molecules. (**D**) Off-target control data. Control data consists of images collected from randomly selected sites at which no target molecule is present. Such sites can be AOIs in which no fluorescent target molecule is visible (e.g., the yellow square in the DNA channel shown in B). Alternatively, control data can be taken from a recording of a separate control sample to which no target molecules were added. Image data in B, C, and D is from Data set A in **Table 1**.

processes in vitro. Examples include cell cycle regulation (**Lu et al., 2015b**), ubiquitination and proteasome-mediated protein degradation (**Lu et al., 2015a**), DNA replication (**Geertsema et al., 2014**; **Ticau et al., 2015**), transcription (**Zhang et al., 2012**; **Friedman and Gelles, 2012**; **Friedman et al., 2013**), micro-RNA regulation (**Salomon et al., 2015**), pre-mRNA splicing (**Shcherbakova et al., 2013**; **Krishnan et al., 2013**; **Warnasooriya and Rueda, 2014**), ribosome assembly (**Kim et al., 2014**), translation (**Wang et al., 2015**; **Tsai et al., 2014**; **O'Leary et al., 2013**), signal recognition particle-nascent protein interaction (**Noriega et al., 2014**), and cytoskeletal regulation (**Smith et al., 2013**; **Breitsprecher et al., 2012**).

**Figure 1A** illustrates an example CoSMoS experiment to measure the interaction kinetics of RNA polymerase II molecules with DNA. In the experiment (**Rosen et al., 2020**), we first measured the locations of individual DNA molecules (the 'targets') tethered to the surface of an observation chamber at low density. Next, a cell extract solution containing fluorescent RNA polymerase II molecules (the 'binders') was added to the solution over the surface and the chamber surface was imaged by total internal reflection fluorescence (TIRF) microscopy. When the binder molecules are freely diffusing in

solution, they are not visible in TIRF. In contrast, when bound to a target, a single binder molecule is detected as a discrete fluorescent spot colocalized with the target position (*Friedman et al., 2006*; *Friedman and Gelles, 2015*).

Effective data analysis is a major challenge in the use of the CoSMoS technique. The basic goal is to acquire information at each time point about whether a binder molecule fluorescence spot is observed at the image position of a target molecule (e.g., whether a colocalized green-dye-labeled RNA polymerase II is observed at the surface location of a blue-dye-labeled DNA spot in *Figure 1B*). Although CoSMoS images are conceptually simple – they consist only of diffraction-limited fluorescent spots collected in several wavelength channels – efficient analysis of the images is inherently challenging. The number of photons emitted by a single fluorophore is limited by fluorophore photobleaching. Consequently, it is desirable to work at the lowest feasible excitation power in order to maximize the duration of experimental recordings and to efficiently capture relevant reaction events. Achieving higher time resolution divides the number of emitted photons between a larger number of images, so that photon shot noise ordinarily dominates the data statistics. Furthermore, the required concentrations of binder molecules can sometimes create significant background noise (*Peng et al., 2018*; *van Oijen, 2011*), even with zero-mode waveguide instruments (*Chen et al., 2014*). These technical difficulties frequently result in CoSMoS images that have low signal-to-noise ratios (SNR), making discrimination of colocalized fluorescence spots from noise a significant challenge. In addition, there are usually non-specific interactions of the binder molecule with the chamber surface, and these artefacts can give rise to both false positive and false negative spot detection (*Friedman and Gelles, 2015*). Together, these defects in analyzing spot colocalization interfere with the interpretation of CoSMoS data to measure reaction thermodynamics and kinetics and to infer molecular mechanisms.

Most CoSMoS spot detection methods are based on integrating the binder fluorescence intensity by summing the pixel values in small regions of the image centered on the location of individual target molecules, and then using crossings of an intensity threshold to score binder molecule arrival and departure, e.g., (*Friedman and Gelles, 2012*; *Shcherbakova et al., 2013*). However, integration discards data about the spatial distribution of intensity that can (and should) be used to distinguish authentic on-target spots from artefacts caused by noise or off-target binding. More recently, improved methods (*Friedman and Gelles, 2015*; *Smith et al., 2019*) were developed that directly analyze TIRF images, using the spatial distribution of binder fluorescence intensity around the target molecule location. All these methods, whether image- or integrated intensity-based, make a binary decision about the presence or absence of a binder spot at the target location. Treating all such binary decisions as equal neglects differences in the confidence of each spot detection decision caused by variations in noise, signal intensity, and non-specific binding. Failure to account for spot confidence decreases the reliability of downstream thermodynamic and kinetic analysis.

In this paper, we describe a qualitatively different Bayesian machine learning method for analysis of CoSMoS data implemented in a computer program, Tapqir (Kazakh: clever, inventive; pronunciation: *tap-keer*). Tapqir analyzes two-dimensional image data, not integrated intensities. Unlike prior methods, our approach is based on an explicit, global causal model for CoSMoS image formation and uses variational Bayesian inference (*Kinz-Thompson et al., 2021*; *Gelman et al., 2013*) to determine the values of model parameters and their associated uncertainties. This model, which we call '*cosmos*', implements time-independent analysis of single-channel (i.e., one-binder) data sets. The *cosmos* model is physics-informed and includes realistic shot noise in fluorescent spots and background, camera noise, the size and shape of spots, and the presence of both target-specific and nonspecific binder molecules in the images. Most importantly, instead of yielding a binary spot-/no-spot determination, the algorithm calculates the probability of a target-specific spot being present at each time point and target location. The calculated probability can then be used in subsequent analyses of the molecular thermodynamics and kinetics. Unlike alternative approaches, Tapqir and *cosmos* do not require subjective threshold settings so they can be used effectively and accurately by non-expert analysts. The program is implemented in the Python-based probabilistic programming language Pyro (*Bingham et al., 2019*), which enables efficient use of graphics processing unit (GPU)-based hardware for rapid parallel processing of data and facilitates future modifications to the model.

## Results

### Data analysis pipeline

The initial steps in CoSMoS data analysis involve preprocessing the data set (*Figure 1B*) to map the spatial relationship between target and binder images, correct for microscope drift (if any) and list the locations of target molecules. Software packages that perform these preprocessing steps are widely available (e.g., *Friedman and Gelles, 2015*; *Smith et al., 2019*).

The input into Tapqir consists of the time sequence of images (*Figure 1B*, right). For colocalization analysis, it is sufficient to consider the image area local to the target molecule. This analyzed area of interest (AOI) needs to be several times the diameter of a diffraction-limited spot to include both the spot and the surrounding background (*Figure 1C*).

In addition to AOIs centered at target molecules, it is useful to also select negative control AOIs from randomly selected sites at which no target molecule is present (*Figure 1B and D*). In Tapqir, such off-target control data is analyzed jointly with on-target data and serves to estimate the background level of target-nonspecific binding.

Once provided with the preprocessing data and image sequence, Tapqir computes for each frame of each AOI the probability, $p$(specific), that a target-specific fluorescence spot is present. The $p$(specific) values that are output can then be used to extract information about the kinetics and thermodynamics of the target-binder interaction.

### Bayesian image classification analysis

Tapqir calculates $p$(specific) values using an objective image classification method built on a rigorous Bayesian statistical approach to the CoSMoS image analysis problem. The Bayesian approach has three components. First, we define a probabilistic model of the CoSMoS images. The probabilistic model, *cosmos*, is a mathematical formalism that describes the AOI images in terms of a set of parameter values. The model is probabilistic in that each parameter is specified to have a probability distribution that defines the likelihood that it can take on particular values. Model parameters describe physically realistic image features such as the characteristic fluorescence spot width. Second, we specify prior distributions for the parameters of the model. These priors embed pre-existing knowledge about the CoSMoS experiment, such as the fact that target-specific spots will be close to the target molecule locations. Third, we infer the values of the model parameters, including $p$(specific), using Bayes' rule (*Bishop, 2006*; *Kinz-Thompson et al., 2021*). The *cosmos* model is 'time-independent', meaning that we ignore the time dimension of the recording – the order of the images does not affect the results.

### Probabilistic image model and parameters

A single AOI image from a CoSMoS data set is a matrix of noisy pixel intensity values. In each image, multiple binder molecule fluorescence spots can be present. *Figure 2A* shows an example image where two spots are present; one spot is located near the target molecule at the center of the image and another is off-target.

The probabilistic model mathematically generates images $D$ as follows. We construct a noise-free AOI image $\mu^I$ as a constant average background intensity $b$ summed with fluorescence spots modeled as 2-D Gaussians $\mu^S$, which accurately approximate the microscope point spread function (*Zhang et al., 2007*; *Figure 2B*). Each 2-D Gaussian is described by parameters integrated intensity $h$, width $w$, and position $(x, y)$. We define $K$ as the maximum number of spots that can be present in a single AOI image. For the data we typically encounter, $K = 2$ is sufficient. Since the spots may be present or not in a particular image, we define the $K = 2$ binary indicators $m_{\text{spot}(1)}$ and $m_{\text{spot}(2)}$. Each indicator can take a value of either 0 denoting spot absence or 1 denoting spot presence.

The resulting mixture model has four possible combinations for $m_{\text{spot}(1)}$ and $m_{\text{spot}(2)}$: (1) a no-spot image that contains only background (*Figure 2B*, top left), (2) a single-spot image that contains the first binder molecule spot superimposed on background (*Figure 2B*, bottom left), (3) a single-spot image that contains the second binder molecule spot superimposed on background (*Figure 2B*, top right), and (4) a two-spot image that contains both binder molecule spots superimposed on background (*Figure 2B*, bottom right).

Among the spots that are present in an AOI image, by assumption at most only one can be target-specific. We use a *state* parameter $z$ to indicate target-specific spot absence ($z = 0$) or presence ($z = 1$) in an AOI image. We also introduce an *index* parameter $\theta$ that identifies which of the spots is the

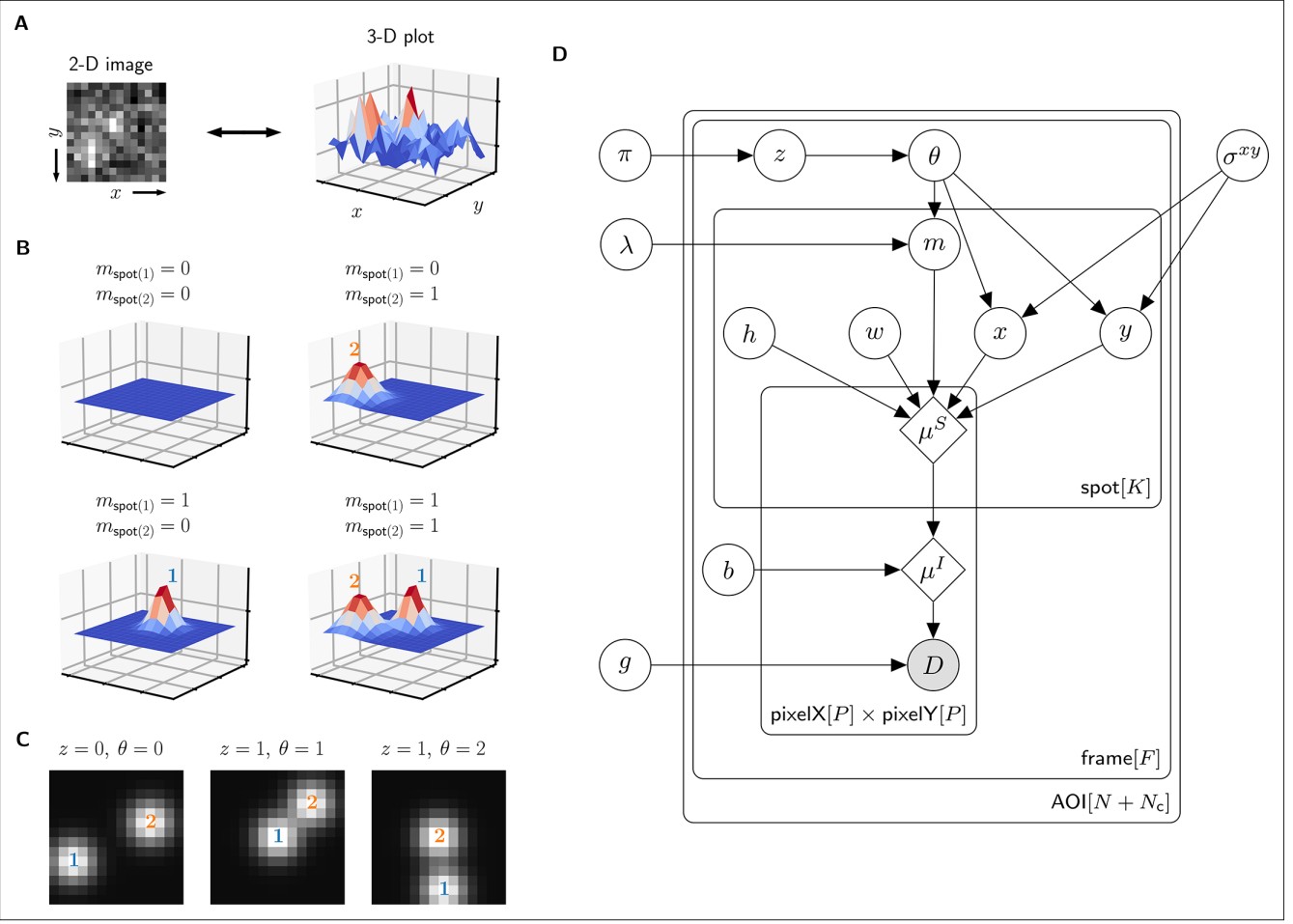

**Figure 2.** Depiction of the *cosmos* probabilistic image model and model parameters. (**A**) Example AOI image (from Data set A in ***Table 1***). The AOI image is a matrix of 14 × 14 pixel intensities which is shown here as both a 2-D grayscale image and as a 3-D intensity plot. The image contains two spots; one is centered at target location (image center) and the other is located off-target. (**B**) Examples of four idealized noise-free image representations ($\mu^I$). Image representations consist of zero, one, or two idealized spots ($\mu^S$) superimposed on a constant background (**b**). Each fluorescent spot is represented as a 2-D Gaussian parameterized by integrated intensity (**h**), width (**w**), and position (**x**, **y**). The presence of spots is encoded in the binary spot existence indicator **m**. (**C**) Simulated idealized images illustrating different values of the target-specific spot state parameter $z$ and index parameter $\theta$. $\theta = 0$ corresponds to a case when no specifically bound molecule is present ($z = 0$); $\theta = 1$ or 2 corresponds to the cases in which specifically bound molecule is present ($z = 1$) and corresponds to spot 1 or 2, respectively. (**D**) Condensed graphical representation of the *cosmos* probabilistic model. Model parameters are depicted as circles and deterministic functions as diamonds. Observed image (**D**) is represented by a shaded circle. Related nodes are connected by edges, with an arrow pointing towards the dependent node (e.g., the shape of each 2-D Gaussian spot $\mu^S$ depends on spot parameters **m**, **h**, **w**, **x**, and **y**). Plates (rounded rectangles) contain entities that are repeated for the number of instances displayed at the bottom-right corner: number of total AOIs ($N + N_c$), frame count (**F**), and maximum number of spots in a single image ($K = 2$). Parameters outside of the plates are global quantities that apply to all frames of all AOIs. A more complete version of the graphical model specifying the relevant probability distributions is given in ***Figure 2—figure supplement 1***.

The online version of this article includes the following figure supplement(s) for figure 2:

**Figure supplement 1.** Extended graphical representation of the generative probabilistic model.

**Figure supplement 2.** Prior distributions for the $x$ and $y$ spot position parameters.

target-specific spot when it is present ($z = 1$) (e.g., ***Figure 2C***, middle and right have $\theta = 1$ and $\theta = 2$, respectively) and equals zero when it is absent ($z = 0$) (e.g., ***Figure 2C***, left). Since the off-target control AOIs by definition contain only non-specific binding, $z = 0$ and $\theta = 0$ for all off-target AOIs.

Finally, to construct realistic noisy AOI images $D$ from the noise-free images $\mu^I$, the model adds intensity-dependent noise to each pixel. For cameras that use charge-coupled device (CCD) or electron-multiplier CCD (EMCCD) sensors, each measured pixel intensity in a single-molecule fluorescence image has a noise contribution from photon counting (shot noise) and can also contain

additional noise arising from electronic amplification (**van Vliet et al., 1998**). The result is a characteristic linear relationship between the noise variance and mean intensity with slope defining the gain $g$. This relationship is used to compute the random pixel noise values (see Materials and methods).

The resulting probabilistic image model can be interpreted as a generative process that produces the observed image data $D$. A graphical representation of the probabilistic relationships in the model is shown in **Figure 2D**. A complete description of the model is given in Materials and methods and **Figure 2—figure supplement 1**.

## Parameter prior distributions

Specifying prior probability distributions for model parameters is essential for Bayesian analysis and allows us to incorporate pre-existing knowledge about the experimental design. For most model parameters, there is no strong prior information so we use uninformative prior distributions (see Materials and methods). However, we have strong expectations for the positions of specific and non-specific binder molecules that can be expressed as prior distributions and used effectively to discriminate between the two. Non-specific binding can occur anywhere on the surface with equal probability and thus has a uniform prior distribution across the AOI image. Target-specific binding, on the other hand, is colocalized with the target molecule and thus has a prior distribution peaked at the AOI center (**Figure 2—figure supplement 2**). The width of this peak, proximity parameter $\sigma^{xy}$, depends on multiple features of the experiment such as the spot localization accuracy and the mapping accuracy between target and binder imaging channels. Prior distributions for parameters $\theta$ and $m$ are defined in terms of the average number of target-specific and target non-specific spots per AOI image, $\pi$ and $\lambda$, respectively. To facilitate convenient use of the algorithm, it is not necessary to pre-specify values of $\sigma^{xy}$, $\pi$, and $\lambda$. Instead, values of these parameters appropriate to a given data set are calculated automatically using a hierarchical Bayesian analysis (see Materials and methods; for hierarchical modeling see Chapter 5 of **Gelman et al., 2013**).

## Bayesian inference and implementation

Tapqir calculates posterior distributions of model parameters conditioned on the observed data by using Bayes' theorem. In particular, Tapqir approximates posterior distributions using a variational inference approach implemented in Pyro (**Bingham et al., 2019**). Complete details of the implementation are given in Materials and methods.

## Tapqir analysis

In initial tests, we used Tapqir to analyze simulated CoSMoS image data with a comparatively high SNR of 3.76 as well as data from the experiment shown in **Figure 1B–D**, which has a lower SNR of 1.61. The simulated data were generated using the same *cosmos* model (**Figure 2D**) that was used for analysis. Tapqir correctly detects fluorescent spots in both simulated and experimental images (compare 'AOI images' and 'Spot-detection' rows in **Figure 3**). The program precisely calculates the position $(x, y)$, intensity $(h)$, and width $(w)$ for each spot and also determines the background intensity $(b)$ for each image without requiring a separate analysis. These parameters confirm the desired behavior of the model and could be used in further calculations. However, the most important output of the analysis is assessment of the presence of target-specific binding. For each AOI image, we calculate $p(\text{specific}) \equiv p(z = 1)$ (**Figure 3**, green), the probability that any target-specific spot is present. Spots determined as likely target-specific ($p(\text{specific}) > 0.5$) are represented as filled circles in the spot detection row of **Figure 3**. For a particular spot to have high $p(\text{specific})$, it must have a high spot probability and be colocalized with the target molecule at the center of the AOI (**Figure 3—figure supplement 1**).

## Tapqir robustly fits experimental data sets with different characteristics

Next, we evaluated how well the model fits data sets encompassing a range of characteristics found in typical CoSMoS experiments. We analyzed four experimental data sets with varying SNR, frequency of target-specific spots, and frequencies of non-specific spots (**Table 1**). We then sampled AOI images from the posterior distributions of parameters (a method known as posterior predictive checking **Gelman et al., 2013**). These posterior predictive simulations accurately reproduce the experimental AOI appearances, recapitulating the noise characteristics and the numbers, intensities, shapes, and

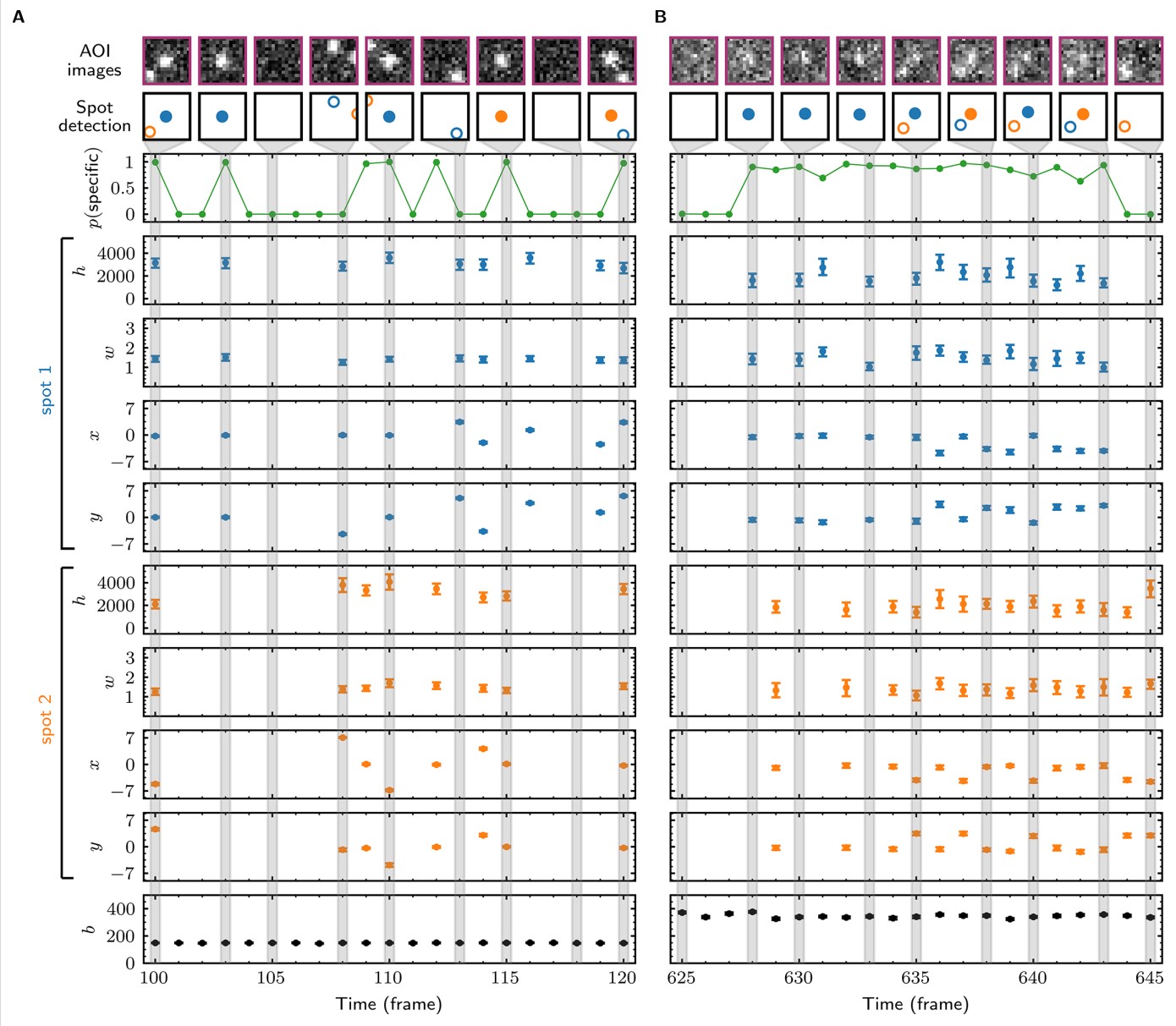

**Figure 3.** Tapqir analysis and inferred model parameters. (**A,B**) Tapqir was applied to simulated data (`lamda0.5` parameter set in *Supplementary file 1*) (**A**) and to experimental data (Data set A in *Table 1*) (**B**). (**A**) and (**B**) each show a short extract from a single target location in the data set. The first row shows AOI images for the subset of frames indicated by gray shaded stripes in the plots; image contrast and offset settings are consistent within each panel. The second row shows the locations of spots determined by Tapqir. Spot numbers 1 (blue) and 2 (orange) are assigned arbitrarily and may change from fame to frame. For clarity, only data for spots with a spot probability $p(m = 1) > 0.5$ are shown. Spots predicted to be target-specific ($p(\theta = k) > 0.5$ for spot $k$) are shown as filled circles. The topmost graphs (green) show the calculated probability that a target-specific spot is present ($p(\text{specific})$) in each frame. Below are the calculated spot intensities ($h$), spot widths ($w$), and locations ($x$, $y$) for spot 1 (blue) and spot 2 (orange), and the AOI background intensities ($b$). Again, for clarity data are only shown for likely spots ($p(m = 1) > 0.5$). Error bars: 95% CI (credible interval) estimated from a sample size of 500. Some error bars are smaller than the points and thus not visible.

The online version of this article includes the following figure supplement(s) for figure 3:

**Figure supplement 1.** Calculated spot probabilities.

**Figure supplement 2.** Reproduction of experimental data by posterior predictive sampling.

**Figure supplement 3.** Tapqir analysis of image data simulated using a broad range of global parameters.

**Figure supplement 4.** Effect of AOI size on analysis of experimental data.

**Table 1.** Experimental data sets.

| Data set size[a] | SNR | π [95% CI] | λ [95% CI] | g [95% CI] | $\sigma^{xy}$ [95% CI] | Compute time |
|---|---|---|---|---|---|---|
| Data set A: Binder, SNAP<sub>f</sub>-tagged S. cerevisiae RNA polymerase II labeled with DY549; Target, transcription template DNA containing 5× Gal4 upstream activating sequences and CYC1 core promoter; Conditions, yeast nuclear extract supplemented with Gal4-VP16 activator and NTPs. From *Rosen et al., 2020*. | | | | | | |
| $N = 331$, $N_c = 526$, $F = 790$ | 1.61 | 0.0951 [0.0936, 0.0966] | 0.2943 [0.2924, 0.2963] | 6.645 [6.643, 6.647] | 0.577 [0.573, 0.580] | 7 h 40 m[b]<br>3 h 50 m[c] |
| Data set B: Binder, 0.1 nM E. coli $\sigma^{54}$ RNA polymerase labeled with Cy3; Target, 852 bp DNA containing the glnALG promoter; Conditions, physiological buffer, no NTPs. From (Fig. 1E) of *Friedman et al., 2013*. | | | | | | |
| $N = 102$, $N_c = 127$, $F = 4407$ | 3.77 | 0.0846 [0.0835, 0.0857] | 0.1575 [0.1569, 0.1583] | 11.861 [11.856, 11.865] | 0.476 [0.474, 0.479] | 7 h 40 m[b] |
| Data set C: Binder, 0.4 nM E. coli $\sigma^{54}$ RNA polymerase labeled with Cy3; Target, 3,591 bp DNA containing the glnALG promoter; Conditions, physiological buffer, no NTPs. From (Fig. 3D) of *Friedman et al., 2013*. | | | | | | |
| $N = 122$, $N_c = 157$, $F = 3855$ | 4.23 | 0.0267 [0.0262, 0.0273] | 0.0876 [0.0869, 0.0883] | 16.777 [16.773, 16.782] | 0.404 [0.399, 0.408] | 9 h 15 m[b] |
| Data set D: Binder, 0.15 nM E. coli Cy3-GreB; Target, reconstituted backtracked EC-6 E. coli transcription elongation complex; Conditions, physiological buffer, no NTPs. Randomly selected subset of data set from *Tetone et al., 2017*. | | | | | | |
| $N = 200$, $N_c = 200$, $F = 5622$ | 3.06 | 0.0038 [0.0036, 0.0039] | 0.0437 [0.0434, 0.0440] | 18.727 [18.724, 18.731] | 0.451 [0.438, 0.463] | 11 h[b] |

*N – number of on-target AOIs, $N_c$ – number of control off-target AOIs, F – number of frames.

[b]Unattended calculation time on an AMD Ryzen Threadripper 2990WX with an Nvidia GeForce RTX 2080Ti GPU using CUDA version 11.5.

[c]Unattended calculation time on an Nvidia Tesla V100-SXM2-16GB GPU using CUDA version 11.2 in a Google Colab Pro account.

locations of spots (*Figure 3—figure supplement 2*, images). The distributions of pixel intensities across the AOI are also closely reproduced (*Figure 3—figure supplement 2*, histograms) confirming that the noise model is accurate. Taken together, these results confirm that the model is rich enough to accurately capture the full range of image characteristics from CoSMoS data sets taken over different experimental conditions. Importantly, all the results on different experimental data sets were obtained using the same model (*Figure 2D*) and the same priors (Materials and methods). No tuning of the algorithm or prior measurement of data-set-specific properties was needed to achieve good fits for all data sets.

## Tapqir accuracy on simulated data with known global parameter values

Next, we evaluated Tapqir's ability to reliably infer the values of global model parameters. To accomplish this, we generated simulated data sets using a wide range of randomized parameter values and then fit the simulated data to the model (*Supplementary file 2*). Fit results show that global model parameters (i.e., average specific spot probability $\pi$, nonspecific binding density $\lambda$, proximity $\sigma^{xy}$, and gain $g$; see *Figure 2D*) are close to the simulated values (*Figure 3—figure supplement 3* and *Supplementary file 2*). This suggests that CoSMoS data contains enough information to reliably infer global model parameters and that the model is not obviously overparameterized.

## Tapqir classification accuracy

Having tested the basic function of the algorithm, we next turned to the key question of how accurately Tapqir can detect target-specific spots in data sets of increasing difficulty.

We first examined the accuracy of target-specific spot detection in simulated data sets with decreasing SNR (*Supplementary file 3*). By eye, spots can be readily discerned at SNR >1 but cannot be clearly seen at SNR <1 (*Figure 4A*). Tapqir gives similar or better performance: if an image contains a target-specific spot, Tapqir correctly assigns it a target-specific spot probability $p(\text{specific})$ that is on average close to one as long as SNR is adequate (i.e., SNR >1) (*Figure 4B*). In contrast, mean $p(\text{specific})$ sharply decreases at SNR <1, consistent with the subjective impression that no spot is recognized under those conditions. In particular, images that contain a target-specific spot are almost always assigned a high $p(\text{specific})$ for high SNR data and almost always assigned low $p(\text{specific})$ for low SNR data (*Figure 4C*, green). At marginal SNR $\approx 1$, these images are assigned a broad distribution of $p(\text{specific})$ values, accurately reflecting the uncertainty in classifying such data. Just as importantly, images with no target-specific spot are almost always assigned $p(\text{specific}) < 0.5$, correctly reflecting the absence of the spot (*Figure 4C*, gray).

Ideally, we want to correctly identify target-specific binding when it occurs but also to avoid incorrectly identifying target-specific binding when it does not occur. To quantify Tapqir's classification accuracy, we next examined binary image classification statistics. Binary classification predictions were obtained by thresholding $p(\text{specific})$ at 0.5. We then calculated two complementary statistics: *recall* and *precision* (*Fawcett, 2006*; *Figure 4D*; see Materials and methods). Recall is defined as the fraction of true target-specific spots that are correctly predicted. Recall is high at high SNR and decreases at lower SNR. Recall is a binary analog of the mean $p(\text{specific})$ for the subset of images containing target-specific spots; as expected the two quantities have similar dependencies on SNR (compare *Figure 4B and D*, black). Precision is the fraction of predicted target-specific spots that are correctly predicted. Precision is near one at all SNR values tested (*Figure 4D*, red); this shows that the algorithm rarely misclassifies an image as containing a target-specific spot when none is present.

In order to quantify the effects of both correctly and incorrectly classified images in a single statistic, we used the binary classification predictions to calculate the Matthews Correlation Coefficient (MCC) (*Matthews, 1975*; see Materials and methods). The MCC is equivalent to the Pearson correlation coefficient between the predicted and true classifications, giving 1 for a perfect match, 0 for a random match, and –1 for complete disagreement. The MCC results (*Figure 4D*, blue) suggest that the overall performance of Tapqir is excellent at SNR $\geq 1$: the program rarely misses target-specific spots that are in reality present and rarely falsely reports a target-specific spot when none is present.

The analyses of *Figure 4B–D* examined Tapqir performance on data in which the rate of target-nonspecific binding is moderate ($\lambda = 0.15$ non-specific spots per AOI image on average). We next examined the effects of increasing the non-specific rate. In particular, we used simulated data (*Supplementary file 1*) with high SNR = 3.76 to test the classification accuracy of Tapqir at different

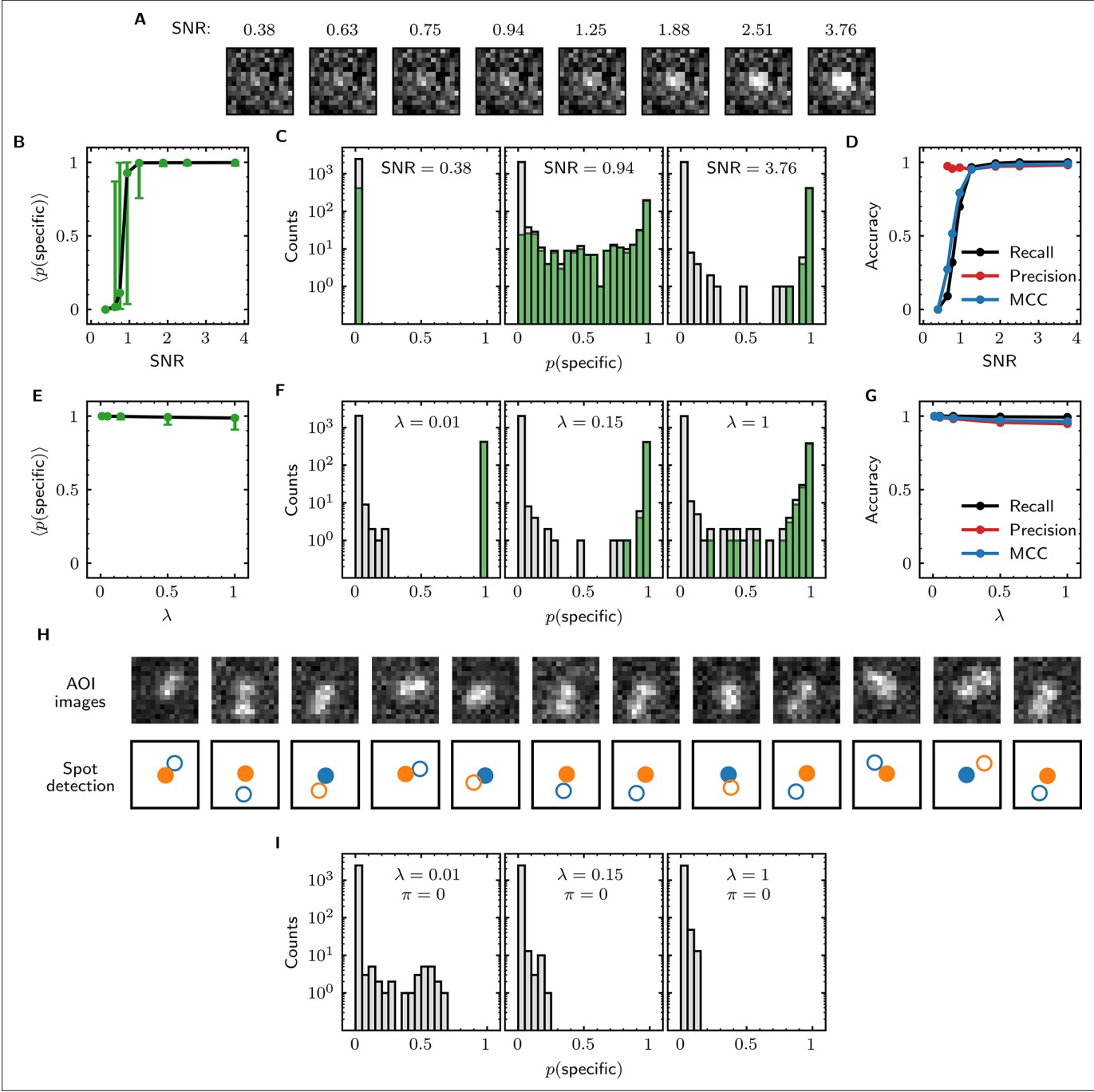

**Figure 4.** Tapqir performance on simulated data with different SNRs or different non-specific binding densities. (**A–D**) Analysis of simulated data over a range of SNR. SNR was varied in the simulations by changing spot intensity $h$ while keeping other parameters constant (***Supplementary file 3***). (**A**) Example images showing the appearance of the same target-specific spot simulated with increasing SNR. (**B**) Mean of Tapqir-calculated target-specific spot probability $p(\text{specific})$ (with 95% CI; see Materials and methods) for the subset of images where target-specific spots are known to be present. (**C**) Histograms of $p(\text{specific})$ for selected simulations with SNR indicated. Data are shown as stacked bars for images known to have (green, 15%) or not have (gray, 85%) target-specific spots. Count is zero for bins where bars are not shown. (**D**) Accuracy of Tapqir image classification with respect to presence/absence of a target-specific spot. Accuracy was assessed by MCC, recall, and precision (see Results and Materials and methods sections). (**E–G**) Same as in (**B–D**) but for the data simulated over a range of non-specific binding densities $\lambda$ at fixed SNR = 3.76 (***Supplementary file 1***). (**H**) Spot recognition in AOI images containing closely spaced target-specific and non-specific spots. Images were selected from the $\lambda = 1$ data set in (**E–G**). AOI images and spot detection are plotted as in ***Figure 3***, with spot numbers 1 (blue) and 2 (orange) assigned arbitrarily and spots predicted to be target-specific shown as filled circles. (**I**) Same as in (**C**) but for the data simulated over a range of non-specific binding densities $\lambda$ with no target-specific binding ($\pi = 0$) (***Supplementary file 4***).

*Figure 4 continued on next page*

*Figure 4 continued*

The online version of this article includes the following figure supplement(s) for figure 4:

**Figure supplement 1.** False negative spot misidentifications by Tapqir and spot-picker method.

non-specific binding densities up to $\lambda = 1$, a value considerably higher than typical of usable experimental data (the experimental data sets in *Table 1* have $\lambda$ ranging from 0.04 to 0.30). In analysis of these data sets, a few images with target-specific spots are misclassified as not having a specific spot ($p$(specific) near zero) or as being ambiguous ($p$(specific) near 0.5) (*Figure 4F*, green bars), and a few images with target-nonspecific spots are misclassified as having specific spot ($p$(specific) near or above 0.5) (*Figure 4F*, gray bars), but these misclassifications only occurred at the unrealistically high $\lambda$ value. Even in the simulation with this highest $\lambda$ value, Tapqir accurately identified target-specific spots (*Figure 4E and F*) and returned excellent binary classification statistics (*Figure 4G*).

A weakness of some existing image-based CoSMoS spot discrimination methods is that target-nonspecific binding adjacent to a target-specific spot can interfere with correctly identifying the latter as target-specific. The very high recall values obtained at $\lambda = 1$ (*Figure 4G*) confirm that there are few such misidentifications by Tapqir even at high non-specific binding densities. This good performance is likely facilitated by the feature of the Tapqir model that explicitly includes the possibility that both a specifically and a non-specifically bound spot may occur simultaneously in the same AOI. Consistent with this interpretation, we see effective detection of the specific and non-specific spots even in example AOIs in which the two spots are so closely spaced that they are not completely resolved (*Figure 4H*). In contrast, tests of existing CoSMoS image classification methods show that images with target-nonspecific spots are prone to misclassification. As discussed previously (*Friedman and Gelles, 2015*), methods based on thresholding of integrated AOI intensities are prone to incorrectly classify target-nonspecific spots as target-specific. Conversely, an existing 'spot-picker' method based on empirical binary classification of 2-D AOI images (*Friedman and Gelles, 2015*) is much more likely than Tapqir to fail to detect target specific spots when there is a nearby non-specific spot (*Figure 4—figure supplement 1*). This contributes to the superior overall performance we see for Tapqir vs. spot-picker on the $\lambda = 1$ data set (recall 0.993 vs 0.919; precision 0.943 vs 0.873; MCC 0.961 vs 0.874).

To further evaluate whether Tapqir is prone to misidentifying target-nonspecific spots as specific, we simulated data sets with no target-specific binding at both low and high non-specific binding densities (*Supplementary file 4*). Analysis of such data (*Figure 4I*) shows that no target-specific binding (i.e., $p$(specific) > 0.6) was detected even under the highest non-specific binding density, demonstrating that Tapqir is robust to false-positive target-specific spot detection even under these extreme conditions.

Since target-nonspecific spots are built into the *cosmos* model, there is no need to choose excessively small AOIs in an attempt to exclude non-specific spots from analysis. We found that reducing AOI size (from 14 × 14 to 6 x 6 pixels) did not appreciably affect analysis accuracy on simulated data, when the width ($w$) of the spots was equal to 1.4 pixels (*Table 2*). In analysis of experimental data, smaller AOI sizes caused occasional changes in calculated $p$(specific) values reflecting apparent missed detection of a few spots (*Figure 3—figure supplement 4*). Out of caution, we therefore used 14 × 14 pixel AOIs routinely, even though the larger AOIs somewhat reduced computation speed (*Table 2* and *Figure 3—figure supplement 4*).

**Table 2.** The effect of AOI size on classification accuracy*.

| AOI dimension[†], P (pixels) | MCC | Compute time[‡] |
|---|---|---|
| 14 | 0.951 | 2 h 10 m |
| 10 | 0.948 | 1 h 25 m |
| 6 | 0.939 | 1 h 20 m |

*Tapqir was applied to the same simulated data set (`height1000` parameter set in *Supplementary file 3*; SNR = 1.25) using different AOI sizes.

[†]The width ($w$) of the simulated spots (one standard deviation of the 2-D Gaussian) is equal to 1.4 pixels.

[‡]Unattended calculation time on an AMD Ryzen Threadripper 2990WX with an Nvidia GeForce RTX 2080Ti GPU using CUDA version 11.5.

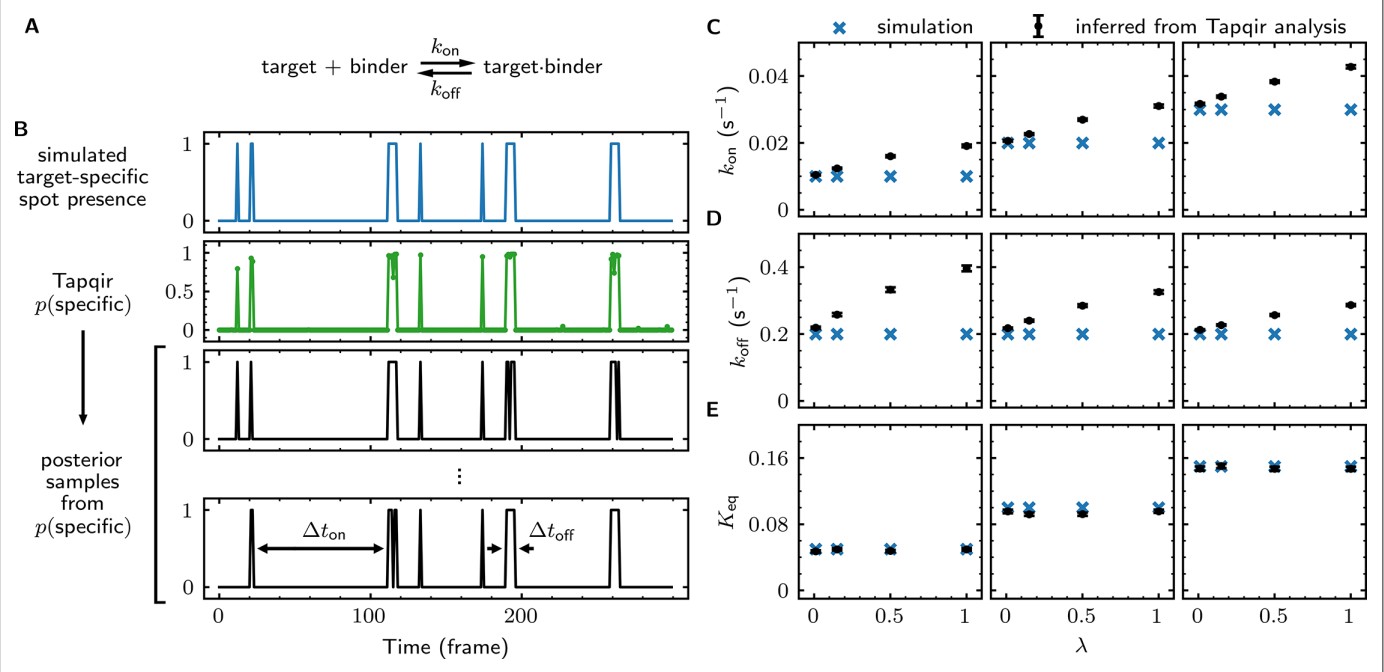

**Figure 5.** Tapqir analysis of association/dissociation kinetics and thermodynamics. (**A**) Chemical scheme for a one-step association/dissociation reaction at equilibrium with pseudo-first-order binding and dissociation rate constants $k_{on}$ and $k_{off}$, respectively. (**B**) A simulation of the reaction in (**A**) and scheme for kinetic analysis of the simulated data with Tapqir. The simulation used SNR = 3.76, $k_{on}$ = 0.02 s$^{-1}$, $k_{off}$ = 0.2 s$^{-1}$, and a high target-nonspecific binding frequency $\lambda$ = 1 (***Supplementary file 5***, data set `kon0.021amda1`). Full dataset consists of 100 AOI locations and 1,000 frames each for on-target data and off-target control data. Shown is a short extract of on-target data from a single AOI location in the simulation. Plots show simulated presence/absence of the target-specific spot (blue) and Tapqir-calculated estimate of corresponding target-specific spot probability $p$(specific) (green). Two thousand binary traces (e.g., black records) were sampled from the $p$(specific) posterior distribution and used to infer $k_{on}$ and $k_{off}$ using a two-state hidden Markov model (HMM) (see Materials and methods). Each sample trace contains well-defined time intervals corresponding to target-specific spot presence and absence (e.g., $\Delta t_{on}$ and $\Delta t_{off}$). (**C,D,E**) Kinetic and equilibrium constants from simulations (***Supplementary file 5***) using a range of $k_{on}$ values and target-nonspecific spot frequencies $\lambda$, with constant $k_{off}$ = 0.2 s$^{-1}$. (**C**) Values of $k_{on}$ used in simulations (blue) and mean values (and 95% CIs, black) inferred by HMM analysis from the 2000 posterior samples. Some error bars are smaller than the points and thus not visible. (**D**) Same as (**C**) but for $k_{off}$. (**E**) Binding equilibrium constants $K_{eq} = k_{on}/k_{off}$ used in simulation (blue) and inferred from Tapqir-calculated $\pi$ as $K_{eq} = \pi/(1 - \pi)$ (black).

## Kinetic and thermodynamic analysis of molecular interactions

The most widespread application of CoSMoS experiments is to measure rate and equilibrium constants for the binding interaction of the target and binder molecules being studied. We next tested whether these constants can be accurately determined using Tapqir-calculated posterior predictions.

We first simulated CoSMoS data sets (***Supplementary file 5***) that reproduced the behavior of a one-step association/dissociation reaction mechanism (***Figure 5A and B***, blue). Simulated data were analyzed with Tapqir yielding $p$(specific) values for each frame (e.g., ***Figure 5B***, green). We wanted to estimate rate constants using the full information contained in the $p$(specific) probabilities, so we did not threshold $p$(specific) for this analysis. Instead, from each single-AOI $p$(specific) time record we constructed a family of binary time records (***Figure 5B***, black) by Monte Carlo sampling according to the $p$(specific) time series. Each family member has well-defined target-specific binder-present and binder-absent intervals $\Delta t_{on}$ and $\Delta t_{off}$, respectively. Each of these time records was then analyzed with a two-state hidden Markov model (HMM) (see Materials and methods), producing a distribution of inferred rate constants from which we calculated mean values and their uncertainties (***Figure 5C and D***). Comparison of the simulated and inferred values shows that both $k_{on}$ and $k_{off}$ rate constants are accurate within 30% at nonspecific binding densities typical of experimental data ($\lambda \leq 0.5$). At higher nonspecific binding densities, rare interruptions caused by false-positive and false-negative spot detection shorten $\Delta t_{on}$ and $\Delta t_{off}$ distributions, leading to moderate systematic overestimation of the association and dissociation rate constants.

From the same simulated data, we calculated the equilibrium constant $K_{eq}$ and its uncertainty. This calculation does not require a time-dependent model and can be obtained directly from the posterior

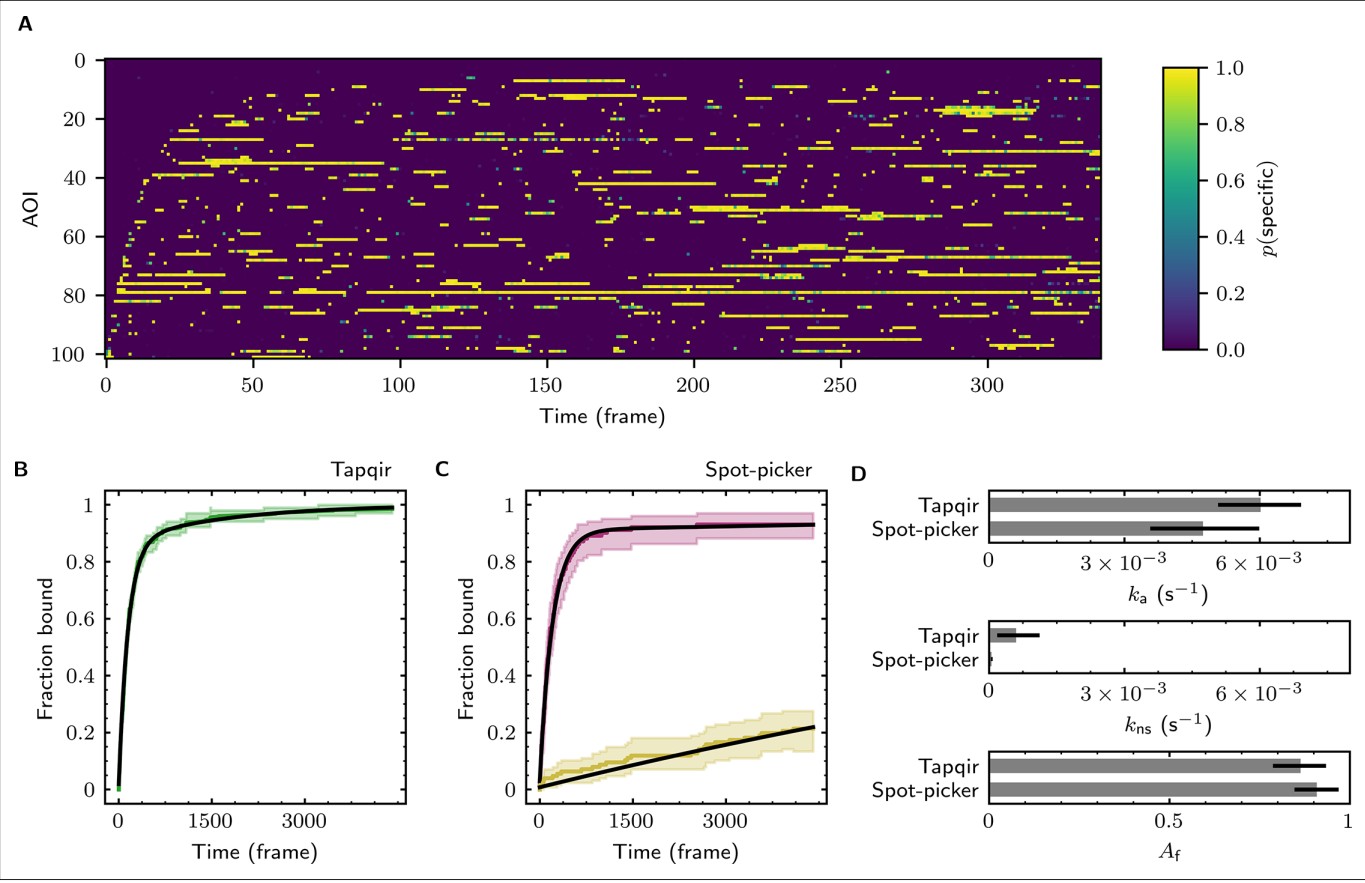

**Figure 6.** Extraction of target-binder association kinetics from example experimental data. Data are from Data set B (SNR = 3.77, $\lambda$ = 0.1575; see *Table 1*). (**A**) Probabilistic rastergram representation of Tapqir-calculated target-specific spot probabilities $p$(**specific**) (color scale). AOIs were ordered by decreasing times-to-first-binding. For clarity, only every thirteenth frame is plotted. (**B**) Time-to-first-binding distribution using Tapqir. Plot shows the cumulative fraction of AOIs that exhibited one or more target-specific binding events by the indicated frame number (green) and fit curve (black). Shading indicates uncertainty. (**C**) Time-to-first-binding distribution using an empirical spot-picker method *Friedman et al., 2013*. The spot-picker method jointly fits first spots observed in off-target control AOIs (yellow) and in on-target AOIs (purple) yielding fit curves (black). (**D**) Values of kinetic parameters $k_a$, $k_{ns}$, and $A_f$ (see text) derived from fits in (**B**) and (**C**). Uncertainties reported in (**B, C, D**) represent 95% credible intervals for Tapqir and 95% confidence intervals for spot-picker (see Materials and methods).

The online version of this article includes the following figure supplement(s) for figure 6:

**Figure supplement 1.** Additional example showing extraction of target-binder association kinetics from experimental data.

**Figure supplement 2.** Additional example showing extraction of target-binder association kinetics from experimental data.

**Figure supplement 3.** Additional example showing extraction of target-binder association kinetics from experimental data.

distribution of the average specific-binding probability $\pi$. The estimated equilibrium constants are highly accurate even at excessively high values of $\lambda$ (*Figure 5E*). The high accuracy results from the fact that equilibrium constant measurements are in general much less affected than kinetic measurements by occasional false positives and false negatives in spot detection.

The forgoing analysis shows that Tapqir can accurately recover kinetic and thermodynamic constants from simulated CoSMoS data. However, experimental CoSMoS data sets can be more diverse. In addition to having different SNR and non-specific binding frequency values, they also may have non-idealities in spot shape (caused by optical aberrations) and in noise (caused by molecular diffusion in and out of the TIRF evanescent field). In order to see if Tapqir analysis is robust to these and other properties of real experimental data, we analyzed several CoSMoS data sets taken from different experimental projects. Analysis of each data set took a few hours of computation time on a GPU-equipped desktop computer or cloud computing service (*Table 1*). We first visualized the results as probabilistic rastergrams (*Figure 6A*, *Figure 6—figure supplement 1A*, *Figure 6—figure supplement 2A*, and *Figure 6—figure supplement 3A*), in which each horizontal line represents the time record from a

single AOI. Unlike the binary spot/no-spot rastergrams in previous studies (e.g., *Friedman et al., 2013*; *Rosen et al., 2020*) we plotted the Tapqir-calculated spot probability $p$(specific) using a color scale. This representation allows a more nuanced understanding of the data. For example, *Figure 6A* reveals that while the long-duration spot detection events typically are assigned a high probability (yellow), some of the shortest duration events have an intermediate $p$(specific) (green) indicating that the assignment of these as target-specific is uncertain.

To demonstrate the utility of Tapqir for kinetic analysis of real experimental data, we measured binder association rate constants in previously published experimental data sets (*Table 1*). We employed our previous strategy (*Friedman and Gelles, 2012*; *Friedman and Gelles, 2015*) of analyzing the duration of the binder-absent intervals that preceded the first binding event. Such time-to-first binding analysis improves the accuracy of association rate constant estimates relative to those obtained by analyzing all $\Delta t_{off}$ values by minimizing the effects of target molecules occupied by photobleached binders, dye blinking and false negative dropouts that occur within a continuous binder dwell interval. To perform a time-to-first-binding analysis using Tapqir, we used the posterior sampling method (as in *Figure 5B*, black records) to determine the initial $\Delta t_{off}$ in each AOI record. These data were fit to a kinetic model (*Friedman and Gelles, 2012*; *Friedman and Gelles, 2015*) in which only a fraction of target molecules $A_f$ were binding competent and which includes both exponential target-specific association with rate constant $k_a$, as well as exponential non-specific association with rate constant $k_{ns}$ (*Figure 6B*, *Figure 6—figure supplement 1B*, *Figure 6—figure supplement 2B*, and *Figure 6—figure supplement 3B*). The Tapqir-derived fits showed excellent agreement with the kinetic model.

To further assess the utility of the Tapqir method, we used experimental data sets and compared the Tapqir association kinetics results with those from the previously published empirical binary 'spot-picker' method (*Friedman and Gelles, 2015*; *Figure 6C*, *Figure 6—figure supplement 1C*, *Figure 6—figure supplement 2C*, and *Figure 6—figure supplement 3C*). The values of the association rate constant $k_a$ obtained using these two methods are in good agreement with each other (*Figure 6D*, *Figure 6—figure supplement 1D*, *Figure 6—figure supplement 2D*, and *Figure 6—figure supplement 3D*). We emphasize that while Tapqir is fully objective, achieving these results with the spot-picker method required optimization by subjective adjustment of spot detection thresholds. We noted some differences between the two methods in the non-specific association rate constants $k_{ns}$. Differences are expected because these parameters are defined differently in the different non-specific binding models used in Tapqir and spot-picker (see Materials and methods).

## Discussion

A broad range of physical processes contribute to the formation of CoSMoS images. These include camera and photon noise, target-specific and non-specific binding, and time- and position-dependent variability in fluorophore imaging and image background. Unlike prior CoSMoS analysis methods, Tapqir considers these aspects of imaging in a single, holistic model. This *cosmos* model explicitly includes the uncertainties due to photon noise, camera gain, and spatial variability in intensity offset. The model also includes the possibility of multiple binder molecule fluorescence spots being present in the vicinity of the target, including both target-specific binding and target-nonspecific interactions of binder molecules with the coverslip surface. This explicit modeling of target-nonspecific spots makes it possible to include off-target control data as a part of the experimental data set. Similarly, all AOIs and frames in the data set are simultaneously fit to the global model in a way that allows for realistic frame-to-frame and AOI-to-AOI variability in image formation caused by variations in laser intensity, fluctuations in background, and other non-idealities. The global analysis based on a single, unified model enables the final results (e.g., kinetic and thermodynamic parameters) to be estimated in a way that is cognizant of the known sources of uncertainty in the data.

Previous approaches to CoSMoS data analysis, including our spot-picker method (*Friedman and Gelles, 2015*), did not employ a holistic modeling approach and instead relied on a multi-step process that includes a separate binary classification step. These prior methods require subjective setting of classification thresholds. Because they are not fully objective, such methods cannot reliably account for uncertainties in spot classification, which compromises error estimates in the analysis pipeline downstream of spot classification. One recent approach (*Smith et al., 2019*; *Smith et al., 2015*), which like spot-picker and Tapqir analyzes 2-D images instead of integrated intensities, used a Bayesian kinetic analysis but a frequentist hypothesis test (a generalized likelihood ratio test) for spot

detection. The frequentist method lacks a key advantage of Tapqir's model-based Bayesian approach that here enables prediction of target-specific spot presence probabilities $p$(specific) for each image, rather than a binary 'spot/no spot' classification. In general, previous approaches in essence assume that spot classifications are correct, and thus the uncertainties in the derived molecular properties (e.g., equilibrium constants) are systematically underestimated because the errors in spot classification, which can be large, are not accounted for. By performing a probabilistic spot classification, Tapqir enables reliable inference of molecular properties, such as thermodynamic and kinetic parameters, and allows statistically well-justified estimation of parameter uncertainties. This more inclusive error estimation likely accounts for the generally larger kinetic parameter error bars obtained from Tapqir compared to those from the existing spot-picker analysis method (*Figure 6*, *Figure 6—figure supplement 1*, *Figure 6—figure supplement 2*, and *Figure 6—figure supplement 3*). Even though existing analysis methods take advantage of subjective tuning by a human analyst, our comparisons show that Tapqir performs at least comparably to (*Figure 6*, *Figure 6—figure supplement 1*, *Figure 6—figure supplement 2*, and *Figure 6—figure supplement 3*) and under some conditions much better than (*Figure 4—figure supplement 1*) the existing spot-picker method.

The Tapqir *cosmos* model includes parameters of mechanistic interest, such as the average probability of target-specific binding, as well as 'nuisance' parameters that are not of primary interest but nevertheless essential for image modeling. In previous image-based methods for CoSMoS analysis (e.g., *Friedman and Gelles, 2015*; *Smith et al., 2019*), nuisance parameters were either measured in separate experiments (e.g., gain was determined from calibration data), set heuristically (e.g., a subjective choice of user-set thresholds for spot intensity and proximity in colocalization detection), or determined at a separate analysis step (e.g., rate of non-specific binding). In contrast, Tapqir directly learns parameters from the full set of experimental data, thus eliminating the need for additional experiments, subjective adjustment of tuning parameters, and post-processing steps.

Bayesian analysis has been used previously to analyze data from single-molecule microscopy experiments (e.g., *Kinz-Thompson et al., 2021* and references cited therein). A key feature of Bayesian analysis is that the extent of prior knowledge of all model parameters is explicitly incorporated. Where appropriate, *cosmos* uses relatively uninformative priors that only weakly specify information about the value of the corresponding parameters. In these cases, *cosmos* mostly infers parameter values from the data. In contrast, some priors are more informative. For example, binder molecule spots near the target molecule are more likely to be target-specific rather than target-nonspecific, so we use this known feature of the experiment by encoding the likely position of target-specific binding as a data-based prior. This tactic effectively enables probabilistic classification of spots as either target-specific or target-nonspecific, which would be difficult using other inference methodologies, while still accommodating data sets with different accuracies of mapping between binder and target channels.

Tapqir is implemented in Pyro, a Python-based probabilistic programming language (PPL) (*Bingham et al., 2019*). Probabilistic programming is a relatively new paradigm in which probabilistic models are expressed in a high-level language that allows easy formulation, modification, and automated inference (*van de Meent et al., 2018*). In this work we focused on developing an image model for colocalization detection in a relatively simple binder-target single-molecule experiment. However, Tapqir can be used with more complex models. For example, the *cosmos* model could be naturally extended to multi-state and multi-color analysis. Furthermore, with the development of more efficient sequential hidden Markov modeling algorithms (*Särkkä and García-Fernández, 2019*; *Obermeyer et al., 2019b*) Tapqir can potentially be extended to directly incorporate kinetic processes, allowing direct inference of kinetic mechanisms and rate constants.

Tapqir is free, open-source software. Tapqir is available at https://github.com/gelles-brandeis/tapqir. The results presented here were obtained using release 1.0 of the program (https://github.com/gelles-brandeis/tapqir/releases/tag/v1.0). The Tapqir documentation, which contains tutorials on program use, is at https://tapqir.readthedocs.io/en/stable/. Source data including Figures, Figure supplements, Supplementary files, manuscript text, and the scripts and data used to generate them are available at https://github.com/ordabayevy/tapqir-overleaf.

```
1:  input: D^raw, x^target,raw, y^target,raw
2:  for all frame[F] do
3:      for all AOI[N + N_c] do
4:          ▷ (P − 1)/2 is an AOI image center
5:          ▷ round function rounds to the closest integer
6:          shiftX = round (x^target,raw − (P − 1)/2)
7:          shiftY = round (y^target,raw − (P − 1)/2)
8:          x^target = x^target,raw − shiftX
9:          y^target = y^target,raw − shiftY
10:         D_{AOI(n),pixelX(i),pixelY(j)} = D^raw_{pixelX(i+shiftX_{AOI(n)}),pixelY(j+shiftY_{AOI(n)})}
        return D, x^target, y^target
```

**Figure 7.** Extraction of AOI images from raw images.

The online version of this article includes the following source data for figure 7:

**Source data 1.** Original text for *Figure 7*.

# Materials and methods
## Notation
In the Materials and methods section, we adopt a mathematical notation for multi-dimensional arrays from the field of machine learning (*Chiang et al., 2021*). The notation uses *named axes* and incorporates implicit broadcasting of arrays when their shapes are different.

## Extracting image data
Raw input data into Tapqir consists of (1) binder channel images ($D^{raw}$), each $W \times H$ pixels in size, for each time point (*Figure 1B*, right) and (2) lists of locations, corrected for microscope drift if necessary (*Friedman and Gelles, 2015*), of target molecules and of off-target control locations (*Friedman and Gelles, 2015*) within the raw images. For simplicity, we use the same notation ($x^{target,raw}$, $y^{target,raw}$) both for target molecule locations and off-target control locations. Tapqir extracts a $P \times P$ AOI around each target and off-target location and returns (1) the extracted data set $D$ consisting of a set of $P \times P$ grayscale images, collected at $N$ on-target AOI sites and $N_c$ off-target AOI sites for a range of $F$ frames (*Figure 1C and D*; *Figure 7*), and (2) new target (and off-target) locations ($x^{target}$, $y^{target}$) adjusted relative to extracted images $D$ where $x^{target}$ and $y^{target}$ both lie within the ($P/2 − 1, P/2$) central range of the image. For the data presented in this article, we used $P = 14$. Cartesian pixel indices ($i$, $j$) are integers but also represent the center point of a pixel on the image plane. While experimental intensity measurements are integers, we treat them as continuous values in our analysis.

## The *cosmos* model
Our intent is to model CoSMoS image data by accounting for the significant physical aspects of image formation, such as photon noise and binding of target-specific and target-nonspecific molecules to the microscope slide surface. A graphical representation of the Tapqir model for CoSMoS data similar to that in *Figure 2D* but including probability distributions and other additional detail is shown in *Figure 2—figure supplement 1*. The corresponding generative model represented as pseudocode is shown in *Figure 8*. All variables with short descriptions and their domains are listed in *Table 3*. Below, we describe the model in detail starting with the observed data and the likelihood function and then proceed with model parameters and their prior distributions.

### Image likelihood
We model the image data $D$ as the sum of a photon-independent offset $\delta$ introduced by the camera and the noisy photon-dependent pixel intensity values $I$:

$$D = \delta + I \qquad (1)$$

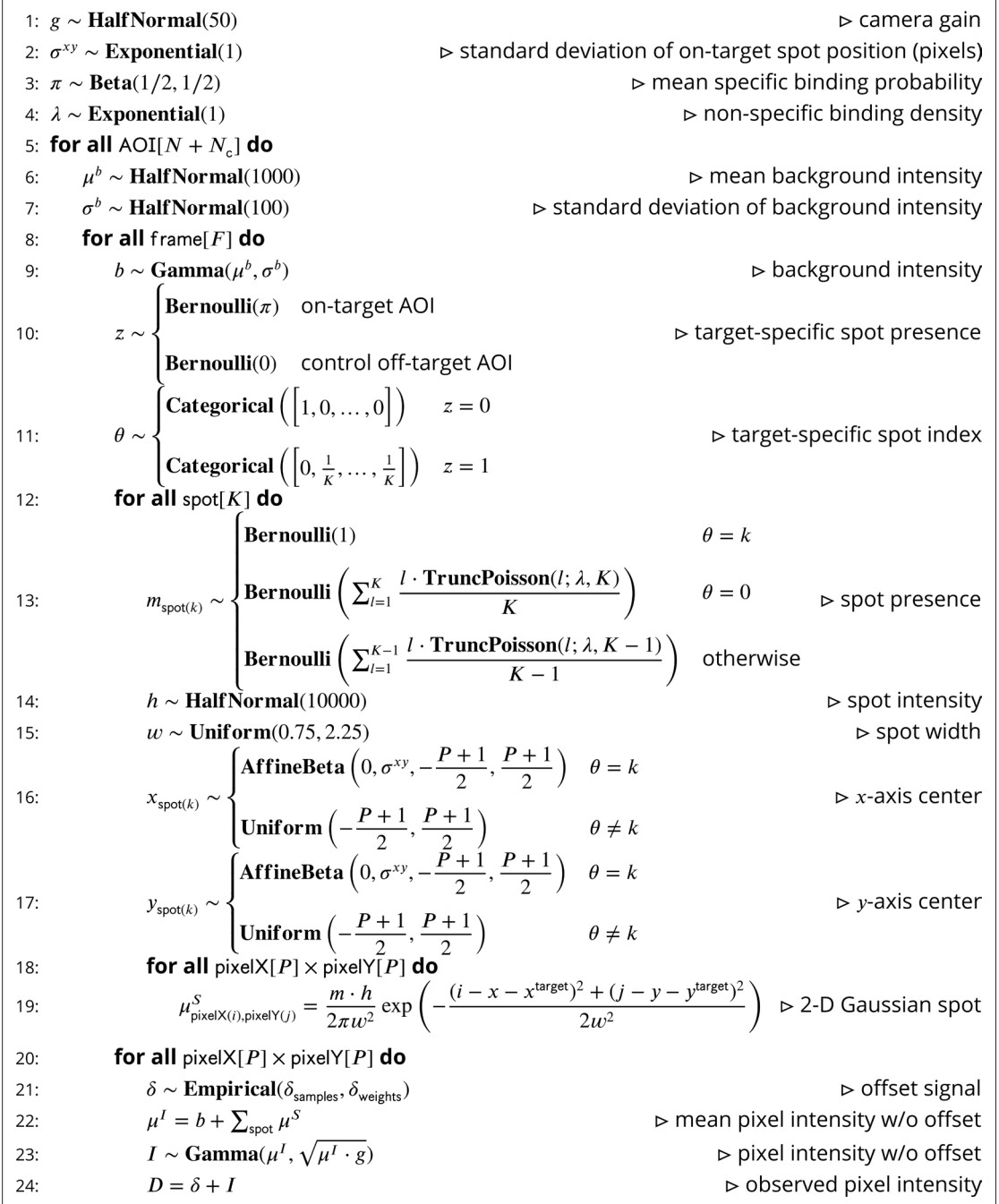

**Figure 8.** Pseudocode representation of *cosmos* model.

The online version of this article includes the following source data for figure 8:

**Source data 1.** Original text for *Figure 8*.

In our model, each pixel in the photon-dependent image $I$ has a variance which is equal to the mean intensity $\mu^I$ of that pixel multiplied by the camera gain $g$, which is the number of camera intensity units per photon. This formulation is appropriate for cameras that use charge-coupled device (CCD) or electron-multiplier CCD (EMCCD) sensors. (The experimental CoSMoS datasets we analyzed (*Table 1*) were collected with EMCCD cameras.) It accounts for both photon shot noise and additional noise introduced by EMCCD camera amplification (*van Vliet et al., 1998*) and is expressed using a continuous Gamma distribution:

**Table 3.** Variables used in the Tapqir model.

| Symbol | Meaning | Domain |
|---|---|---|
| $K$ | Maximum number of spots per image | $\mathbb{N}$ |
| $N$ | Number of on-target AOIs | $\mathbb{N}$ |
| $N_c$ | Number of off-target control AOIs | $\mathbb{N}$ |
| $F$ | Number of frames | $\mathbb{N}$ |
| $P$ | Size of the AOI image in pixels | $\mathbb{N}$ |
| $g$ | Camera gain | $\mathbb{R}_{>0}$ |
| $\sigma^{xy}$ | Proximity | $(0, (P+1)/\sqrt{12})$ |
| $\pi$ | Average target-specific binding probability | $[0, 1]$ |
| $\lambda$ | Target-nonspecific binding density | $\mathbb{R}_{>0}$ |
| $\mu^b$ | Mean background intensity across AOI | $\mathbb{R}_{>0}^{\text{AOI}[N]}$ |
| $\sigma^b$ | Standard deviation of background intensity across AOI | $\mathbb{R}_{>0}^{\text{AOI}[N]}$ |
| $b$ | Background intensity | $\mathbb{R}_{>0}^{\text{AOI}[N]\times\text{frame}[F]}$ |
| $z$ | Target-specific spot presence | $\{0, 1\}^{\text{AOI}[N]\times\text{frame}[F]}$ |
| $\theta$ | Target-specific spot index | $\{0, 1, \ldots, K\}^{\text{AOI}[N]\times\text{frame}[F]}$ |
| $m$ | Spot presence indicator | $\{0, 1\}^{\text{spot}[K]\times\text{AOI}[N]\times\text{frame}[F]}$ |
| $h$ | Integrated spot intensity | $\mathbb{R}_{>0}^{\text{spot}[K]\times\text{AOI}[N]\times\text{frame}[F]}$ |
| $w$ | Spot width | $[0.75, 2.25]^{\text{spot}[K]\times\text{AOI}[N]\times\text{frame}[F]}$ |
| $x$ | Center of the spot on the $x$-axis | $\mathbb{R}^{\text{spot}[K]\times\text{AOI}[N]\times\text{frame}[F]}$ |
| $y$ | Center of the spot on the $y$-axis | $\mathbb{R}^{\text{spot}[K]\times\text{AOI}[N]\times\text{frame}[F]}$ |
| $\mu^S$ | 2-D Gaussian spot | $\mathbb{R}_{>0}^{\text{spot}[K]\times\text{AOI}[N]\times\text{frame}[F]\times\text{pixelX}[P]\times\text{pixelY}[P]}$ |
| $\mu^I$ | Ideal image w/o offset | $\mathbb{R}_{>0}^{\text{AOI}[N]\times\text{frame}[F]\times\text{pixelX}[P]\times\text{pixelY}[P]}$ |
| $\delta$ | Offset signal | $\mathbb{R}_{>0}^{\text{AOI}[N]\times\text{frame}[F]\times\text{pixelX}[P]\times\text{pixelY}[P]}$ |
| $I$ | Observed image w/o offset signal | $\mathbb{R}_{>0}^{\text{AOI}[N]\times\text{frame}[F]\times\text{pixelX}[P]\times\text{pixelY}[P]}$ |
| $D$ | Observed image ($I + \delta$) | $\mathbb{R}_{>0}^{\text{AOI}[N]\times\text{frame}[F]\times\text{pixelX}[P]\times\text{pixelY}[P]}$ |
| $x^{\text{target}}$ | Target molecule position on the $x$-axis | $[P/2 - 1, P/2]^{\text{AOI}[N]\times\text{frame}[F]}$ |
| $y^{\text{target}}$ | Target molecule position on the $y$-axis | $[P/2 - 1, P/2]^{\text{AOI}[N]\times\text{frame}[F]}$ |
| $i$ | Pixel index on the $x$-axis | $\{0, \ldots, (P-1)\}^{\text{pixelX}[P]}$ |
| $j$ | Pixel index on the $y$-axis | $\{0, \ldots, (P-1)\}^{\text{pixelX}[P]}$ |
| $W$ | Width of the raw microscope images in pixels | $\mathbb{N}$ |
| $H$ | Height of the raw microscope image in pixels | $\mathbb{N}$ |
| $D^{\text{raw}}$ | Raw microscope images | $\mathbb{R}_{>0}^{\text{frame}[F]\times\text{pixelX}[H]\times\text{pixelY}[W]}$ |

*Table 3 continued on next page*

**Table 3 continued**

| Symbol | Meaning | Domain |
|---|---|---|
| $x^{\text{target,raw}}$ | Target molecule position in raw images on the $x$-axis | $[-0.5, H - 0.5]^{\text{AOI}[N] \times \text{frame}[F]}$ |
| $y^{\text{target,raw}}$ | Target molecule position in raw images on the $y$-axis | $[-0.5, W - 0.5]^{\text{AOI}[N] \times \text{frame}[F]}$ |

$$I \sim \mathbf{Gamma}(\mu^I, \sqrt{\mu^I \cdot g}) \tag{2}$$

The Gamma distribution was chosen because we found it to effectively model the image noise, which includes both Poissonian (shot noise) and non-Poissonian contributions. The Gamma distribution used here is parameterized by its mean and standard deviation. The functional forms of the Gamma distribution and all other distributions we use in this work are given in **Table 4**.

A competing camera technology based on scientific complementary metal-oxide semiconductor (sCMOS) sensors produces images that have also successfully been modeled as having a combination of Poissonian and non-Poissonian (Gaussian, in this case) noise sources. However, sCMOS images have noise characteristics that are considerably more complicated than CCD/EMCCD images, because every pixel has its own characteristic intensity offset, Gaussian noise variance, and amplification gain. Additional validation will be required to determine whether the existing *cosmos* model requires modification or inclusion of additional prior information (e.g., pixel-by-pixel calibration data as in **Huang et al., 2013**) to optimize its performance with sCMOS CoSMoS data.

## Image model

The idealized noise-free image $\mu^I$ is represented as the sum of a background intensity $b$ and the intensities from fluorescence spots modeled as 2-D Gaussians $\mu^S$:

**Table 4.** Probability distributions used in the model.

| Distribution | PDF |
|---|---|
| $x \sim \mathbf{AffineBeta}(\mu, \nu, a, b)$ | $\dfrac{y^{\alpha-1}(1-y)^{\beta-1}}{\mathrm{B}(\alpha, \beta)}$ where $\alpha = \dfrac{\nu(\mu - a)}{b - a}$, $\beta = \dfrac{\nu(b - \mu)}{b - a}$, and $y = \dfrac{x - a}{b - a}$ |
| $x \sim \mathbf{Bernoulli}(\pi)$ | $\pi^x (1 - \pi)^{1-x}$ |
| $x \sim \mathbf{Beta}(\alpha, \beta)$ | $\dfrac{x^{\alpha-1}(1-x)^{\beta-1}}{\mathrm{B}(\alpha, \beta)}$ |
| $x \sim \mathbf{Categorical}(p)$ | $\prod_{i=1}^{k} p_i^{[x=i]}$ |
| $x \sim \mathbf{Empirical}(z, p)$ | $\prod_{i=1}^{k} p_i^{[x=z_i]}$ |
| $x \sim \mathbf{Exponential}(\lambda)$ | $\lambda e^{-\lambda x}$ |
| $x \sim \mathbf{Gamma}(\mu, \sigma)$ | $\dfrac{\beta^{\alpha}}{\Gamma(\alpha)} x^{\alpha-1} e^{-\beta x}$ where $\alpha = \dfrac{\mu^2}{\sigma^2}$ and $\beta = \dfrac{\mu}{\sigma^2}$ |
| $x \sim \mathbf{HalfNormal}(\sigma)$ | $\dfrac{\sqrt{2}}{\sigma\sqrt{\pi}} \exp\left(-\dfrac{x^2}{2\sigma^2}\right)$ for $x > 0$ |
| $k \sim \mathbf{TruncPoisson}(\lambda, K)$ | $\begin{cases} 1 - e^{-\lambda} \sum_{i=0}^{K-1} \dfrac{\lambda^i}{i!} & \text{if } k = K \\ \dfrac{\lambda^k e^{-\lambda}}{k!} & \text{otherwise} \end{cases}$ |
| $x \sim \mathbf{Uniform}(a, b)$ | $\dfrac{1}{b - a}$ for $x \in [a, b]$ |

$$\mu^I = b + \sum_{\text{spot}} \mu^S \tag{3}$$

For simplicity, we allow at most $K$ number of spots in each frame of each AOI. (In this article, we always use $K$ equal to 2.) The presence of a given spot in the image is encoded in the binary spot existence parameter $m$, where $m = 1$ when the corresponding spot is present and $m = 0$ when it is absent.

The intensities for a 2-D Gaussian spot at each pixel coordinate $(i, j)$ is given by:

$$\mu^S_{\text{pixelX}(i),\text{pixelY}(j)} = \frac{m \cdot h}{2\pi w^2} \exp\left(-\frac{(i - x - x^{\text{target}})^2 + (j - y - y^{\text{target}})^2}{2w^2}\right) \tag{4}$$

with spot parameters total integrated intensity $h$, width $w$, and center $(x, y)$ relative to the target (or off-target control) location ($x^{\text{target}}$, $y^{\text{target}}$).

Our primary interest is whether a target-specific spot is absent or present in a given AOI. We encode this information using a binary *state* parameter $z$ with 0 and 1 denoting target-specific spot absence and presence, respectively. To indicate which of the $K$ spots is target-specific, we use the *index* parameter $\theta$ which ranges from 0 to $K$. When a target-specific spot is present ($z = 1$), $\theta \in \{1, \dots, K\}$ specifies the index of the target-specific spot, while $\theta = 0$ indicates that no target-specific spot is present ($z = 0$). For example, $\{m_{\text{spot}(1)} = 1, m_{\text{spot}(2)} = 1, z = 1, \theta = 2\}$ means that both spots are present and spot 2 is target-specific. A combination like $\{m_{\text{spot}(1)} = 0, m_{\text{spot}(2)} = 1, z = 1, \theta = 1\}$ is impossible (i.e., has zero probability) since spot 1 cannot be absent and target-specific at the same time. For off-target control data, in which no spots are target-specific by definition, $z$ and $\theta$ are always set to zero.

### Prior distributions

The prior distributions for the model parameters are summarized in *Figure 2—figure supplement 1* and detailed below. Unless otherwise indicated we assume largely uninformative priors (such as the Half-Normal distribution with large mean).

Background intensity $b$ follows a Gamma distribution:

$$b \sim \textbf{Gamma}(\mu^b, \sigma^b) \tag{5}$$

*where* the mean $\mu^b \in \mathbb{R}^{\text{AOI}[N]}_{>0}$ and standard deviation $\sigma^b \in \mathbb{R}^{\text{AOI}[N]}_{>0}$ of the background intensity describe the irregularity in the background intensity in time and across the field of view of the microscope. Priors for $\mu^b$ and $\sigma^b$ are uninformative:

$$\mu^b \sim \textbf{HalfNormal}(1000) \tag{6a}$$

$$\sigma^b \sim \textbf{HalfNormal}(100) \tag{6b}$$

The target-specific presence parameter $z$ has a Bernoulli prior parameterized by the average target-specific binding probability $\pi$ for on-target AOIs and zero probability for control off-target AOIs:

$$z \sim \begin{cases} \textbf{Bernoulli}(\pi) & \text{on-target AOI} \\ 0 & \text{control off-target AOI} \end{cases} \tag{7}$$

---

**Table 5.** The effect of mapping precision on classification accuracy*.

| $\sigma^{xy}$(true) | $\sigma^{xy}$(fit) [95% CI] | MCC | $\sigma^{xy}$ Prior |
|---|---|---|---|
| 0.2 | 0.21 [0.20, 0.22] | 0.989 | **Exponential**(1) |
| 1 | 0.96 [0.90, 1.02] | 0.939 | **Exponential**(1) |
| 1.5 | 1.49 [1.40, 1.59] | 0.890 | **Exponential**(1) |
| 2 | 1.96 [1.84, 2.09] | 0.834 | **Exponential**(1) |
| 2 | 1.97 [1.84, 2.09] | 0.834 | **Uniform**$(0, (P + 1)/\sqrt{12})$ |

*Data were simulated over a range of proximity parameter $\sigma^{xy}$ values at fixed $\pi = 0.15$ and $\lambda = 0.15$ (*Supplementary file 6*).

---

The prior distribution for the index of the target-specific spot $\theta$ is conditional on $z$. When no specifically bound spot is present (i.e., $z = 0$), $\theta$ always equals 0. Since spot indices are arbitrarily assigned, when the target-specific spot is present (i.e., $z = 1$) $\theta$ can take any value between 1 and $K$ with equal probability. We represent the prior for $\theta$ as a Categorical distribution of the following form:

$$\theta \sim \begin{cases} 0 & z = 0 \\ \mathbf{Categorical}\left(\left[0, \frac{1}{K}, \ldots, \frac{1}{K}\right]\right) & z = 1 \end{cases} \tag{8}$$

The average target-specific binding probability $\pi$ has an uninformative Jeffreys prior (**Gelman et al., 2013**) given by a Beta distribution:

$$\pi \sim \mathbf{Beta}(1/2, 1/2) \tag{9}$$

The prior distribution for the spot presence indicator $m$ is conditional on $\theta$. When $\theta$ corresponds to spot index $k$, i.e., $\theta = k$, then $m_{\mathrm{spot}(k)} = 1$. When $\theta$ does not correspond to a spot index $k$, that is, $\theta \neq k$, then either spot $k$ is target-nonspecific or a spot corresponding to $k$ does not exist. Consequently, for $\theta \neq k$ we assign $m_{\mathrm{spot}(k)}$ to either 0 or 1 with a probability dependent on the non-specific binding density $\lambda \in \mathbb{R}_{>0}$:

$$m_{\mathrm{spot}(k)} \sim \begin{cases} 1 & \theta = k \\ \mathbf{Bernoulli}\left(\sum_{l=1}^{K} \frac{l \cdot \mathbf{TruncPoisson}(l; \lambda, K)}{K}\right) & \theta = 0 \\ \mathbf{Bernoulli}\left(\sum_{l=1}^{K-1} \frac{l \cdot \mathbf{TruncPoisson}(l; \lambda, K-1)}{K-1}\right) & \text{otherwise} \end{cases} \tag{10}$$

The mean non-specific binding density $\lambda$ is expected to be much less than two non-specifically bound spots per frame per AOI; therefore, we use an Exponential prior of the form

$$\lambda \sim \mathbf{Exponential}(1) \tag{11}$$

The prior distribution for the integrated spot intensity $h$ is chosen to fall off at a value much greater than typical spot intensity values

$$h \sim \mathbf{HalfNormal}(10000) \tag{12}$$

In CoSMoS experiments, the microscope/camera hardware is typically designed to set the width $w$ of fluorescence spots to a typical value in the range of 1–2 pixels (**Ober et al., 2015**). We use a Uniform prior confined to the range between 0.75 and 2.25 pixels:

$$w \sim \mathbf{Uniform}(0.75, 2.25) \tag{13}$$

Priors for spot position $(x, y)$ depend on whether the spot represents target-specific or non-specific binding. Non-specific binding to the microscope slide surface can occur anywhere within the image and therefore has a uniform distribution (**Figure 2—figure supplement 2**, red). Spot centers may fall slightly outside the AOI image yet still affect pixel intensities within the AOI. Therefore the range for $(x, y)$ is extended one pixel wider than the size of the image, which allows a spot center to fall slightly beyond the AOI boundary.

In contrast to non-specifically bound molecules, specifically bound molecules are colocalized with the target molecule with a precision that can be better than one pixel and that depends on various factors including the microscope point-spread function and magnification, accuracy of registration between binder and target image channels, and accuracy of drift correction. For target-specific binding, we use an Affine-Beta prior with zero mean position relative to the target molecule location $(x^{\mathrm{target}}, y^{\mathrm{target}})$, and a 'proximity' parameter $\sigma^{xy}$ which is the standard deviation of the Affine-Beta distribution (**Figure 2—figure supplement 2**, green). We chose the Affine-Beta distribution because it models a continuous parameter defined on a bounded interval.

$$x_{\mathrm{spot}(k)}, y_{\mathrm{spot}(k)} \sim \begin{cases} \mathbf{AffineBeta}\left(0, \sigma^{xy}, -\frac{P+1}{2}, \frac{P+1}{2}\right) & \theta = k \text{ (target-specific)} \\ \mathbf{Uniform}\left(-\frac{P+1}{2}, \frac{P+1}{2}\right) & \theta \neq k \text{ (target-nonspecific)} \end{cases} \tag{14}$$

We give the proximity parameter $\sigma^{xy}$ a diffuse prior, an Exponential with a characteristic width of one pixel:

$$\sigma^{xy} \sim \textbf{Exponential}(1) \tag{15}$$

Tests on data simulated with increasing proximity parameter values $\sigma^{xy}$ (true) (i.e., with decreasing precision of spatial mapping between the binder and target image channels) confirm that the *cosmos* model accurately learns $\sigma^{xy}$ (fit) from the data (*Figure 3—figure supplement 3D*; *Table 5*). This was the case even if we substituted a less-informative $\sigma^{xy}$ prior (Uniform vs. Exponential; *Table 5*).

The CoSMoS technique is premised on colocalization of the binder spots with the known location of the target molecule. Consequently, for any analysis method, classification accuracy declines when the images in the target and binder channels are less accurately mapped. For the Tapqir *cosmos* model, low mapping precision has little effect on classification accuracy at typical non-specific binding densities ($\lambda = 0.15$; see MCC values in *Table 5*).

Gain $g$ depends on the settings of the amplifier and electron multiplier (if present) in the camera. It has a positive value and is typically in the range between 5 and 50. We use a Half-Normal prior with a broad distribution encompassing this range:

$$g \sim \textbf{HalfNormal}(50) \tag{16}$$

The prior distribution for the offset signal $\delta$ is empirically measured from the output of camera sensor regions that are masked from incoming photons. Collected data from these pixels are transformed into a density histogram with intensity step size of 1. The resulting histogram typically has a long right hand tail of low density. For computational efficiency, we shorten this tail by binning together pixel intensity values from the upper 0.5% percentile. Since $D = \delta + I$ (*Equation 1*) and photon-dependent intensity $I$ is positive, all $D$ values have to be larger than the smallest offset intensity value. If that is not the case we add a single value $\min(D) - 1$ to the offset empirical distribution which has a negligible effect on the distribution. Bin values $\delta_{\text{samples}}$ and their weights $\delta_{\text{weights}}$ are used to construct an Empirical prior:

$$\delta \sim \textbf{Empirical}(\delta_{\text{samples}}, \delta_{\text{weights}}) \tag{17}$$

All simulated and experimental data sets in this work were analyzed using the prior distributions and hyperparameter values given above, which are compatible with a broad range of experimental conditions (*Table 1*). Many of the priors are uninformative and we anticipate that these will work well with images taken on variety of microscope hardware. However, it is possible that highly atypical microscope designs (e.g., those with effective magnifications that are sub-optimal for CoSMoS) might require adjustment of some fixed hyperparameters and distributions (those in *Eqs. 6a, 6b, 11, 12, 13, 15, and 16*). For example, if the microscope point spread function is more than 2 pixels wide, it may be necessary to increase the range of the $w$ prior in *Eq. 13*. The Tapqir documentation (https://tapqir.readthedocs.io/en/stable/) gives instructions for changing the hyperparameters.

## Joint distribution

The joint distribution of the data and all parameters is the fundamental distribution necessary to perform a Bayesian analysis. Let $\phi$ be the set of all model parameters. The joint distribution can be expressed in a factorized form:

$$p(D, \phi) = p(g)p(\sigma^{xy})p(\pi)p(\lambda) \prod_{\text{AOI}} \left[ p(\mu^b)p(\sigma^b) \prod_{\text{frame}} \left[ p(b \mid \mu^b, \sigma^b)p(z \mid \pi)p(\theta \mid z) \cdot \right. \right.$$

$$\left. \left. \prod_{\text{spot}} \left[ p(m \mid \theta, \lambda)p(h)p(w)p(x \mid \sigma^{xy}, \theta)p(y \mid \sigma^{xy}, \theta) \right] \prod_{\substack{\text{pixelX} \\ \text{pixelY}}} p(\delta)p(D \mid \mu^I, g, \delta) \right] \right] \tag{18}$$

The Tapqir generative model is a stochastic function that describes a properly normalized joint distribution for the data and all parameters (*Figure 8*). In Pyro this is called 'the model'.

## Inference

For a Bayesian analysis, we want to obtain the posterior distribution for parameters $\phi$ given the observed data $D$. There are three discrete parameters $z$, $\theta$, and $\delta$ that can be marginalized out exactly so that they do not appear expilicty in either the joint posterior distribution or the likelihood function. Computationally efficient marginalization is implemented using Pyro's enumeration strategy (**Obermeyer et al., 2019a**) and KeOps' kernel operations on the GPU without memory overflows (**Charlier et al., 2021**). Let $\phi' = \phi - \{z, \theta, \delta\}$ be the rest of the parameters. We obtain posterior distributions of $\phi'$ using Bayes' rule:

$$p(\phi'|D) = \frac{\sum_{z,\theta,\delta} p(D, \phi)}{\int_\phi p(D, \phi)d\phi} = \frac{p(D, \phi')}{\int_\phi p(D, \phi)d\phi} = \frac{p(D|\phi')p(\phi')}{\int_\phi p(D, \phi)d\phi} \tag{19}$$

Note that the integral in the denominator of this expression is necessary to calculate the posterior distribution, but it is usually analytically intractable. However, variational inference provides a robust method to approximate the posterior distribution $p(\phi'|D)$ with a parameterized variational distribution $q(\phi')$ (**Bishop, 2006**).

$$p(\phi'|D) \simeq q(\phi') \tag{20}$$

$q(\phi')$ has the following factorization:

$$q(\phi') = q(g)q(\sigma^{xy})q(\pi)q(\lambda) \cdot$$
$$\prod_{\text{AOI}}\left[q(\mu^b)q(\sigma^b)\prod_{\text{frame}}\left[q(b)\prod_{\text{spot}}\left[q(m)q(h|m)q(w|m)q(x|m)q(y|m)\right]\right]\right] \tag{21}$$

The variational distribution $q(\phi')$ is provided as pseudocode for a generative stochastic function (**Figure 9**). In Pyro this is called 'the guide'. Variational inference is sensitive to initial values of variational parameters. In **Figure 9**, step 1 we provide the initial values of variational parameters used in our analyses.

## Calculation of spot probabilities

Variational inference directly optimizes $q(m) \equiv m_{\text{prob}}$ (see **Eq. 21** and **Figure 9**), which approximates $p(m|D)$. To obtain the marginal posterior probabilities $p(z, \theta|D)$, we use a Monte Carlo sampling method:

$$
\begin{aligned}
p(z, \theta|D) &= \int_{\phi'} p(z, \theta, \phi'|D)d\phi' \\
&= \int_{\phi'} p(z, \theta|\phi', D)p(\phi'|D)d\phi' \\
&= \int_{\phi'} p(z, \theta|\phi', D)p(\phi'|D)d\phi' \\
&= \int_{\phi'} \frac{p(z, \theta, \phi', D)}{\sum_{z,\theta} p(z, \theta, \phi', D)}p(\phi'|D)d\phi' \\
&\simeq \frac{1}{S}\sum_{s=1}^{S} \frac{p(z, \theta, \phi'_s, D)}{\sum_{z,\theta} p(z, \theta, \phi'_s, D)} \quad \text{where} \quad \phi'_s \sim q(\phi')
\end{aligned}
\tag{22}
$$

In our calculations, we used $S = 25$ as the number of Monte Carlo samples. Marginal probabilities $p(z|D)$ and $p(\theta|D)$ are calculated as:

$$p(z|D) = \sum_{\theta} p(z, \theta|D) \tag{23a}$$

$$p(\theta|D) = \sum_{z} p(z, \theta|D) \tag{23b}$$

The probability, $p(\text{specific})$, that a target-specific fluorescence spot is present in a given image by definition is:

1: **Variational parameter initializations** {initial value,　constraint}:

$g_{\text{mean}} \leftarrow \{5, \quad \mathbb{R}_{>0}\}; g_{\text{beta}} \leftarrow \{100, \quad \mathbb{R}_{>0}\}$

$\sigma^{xy}_{\text{mean}} \leftarrow \{0, \quad (0, (P+1)/\sqrt{12})\}; \sigma^{xy}_{\text{beta}} \leftarrow \{100, \quad \mathbb{R}_{>2}\}$

$\pi_{\text{mean}} \leftarrow \{0.5, \quad [0,1]\}; \pi_{\text{size}} \leftarrow \{2, \quad \mathbb{R}_{>2}\}$

$\lambda_{\text{mean}} \leftarrow \{0.5, \quad \mathbb{R}_{>0}\}; \lambda_{\text{beta}} \leftarrow \{100, \quad \mathbb{R}_{>0}\}$

$\mu^b_{\text{mean}} \leftarrow \{\text{mean}(D)^{\text{AOI}[N]}, \quad \mathbb{R}_{>0}\}; \sigma^b_{\text{mean}} \leftarrow \{1^{\text{AOI}[N]}, \quad \mathbb{R}_{>0}\}$

$b_{\text{mean}} \leftarrow \{\text{mean}(D)^{\text{AOI}[N] \times \text{frame}[F]}, \quad \mathbb{R}_{>0}\}$

$b_{\text{beta}} \leftarrow \{1^{\text{AOI}[N] \times \text{frame}[F]}, \quad \mathbb{R}_{>0}\}$

$m_{\text{prob}} \leftarrow \{0.5^{\text{spot}[K] \times \text{AOI}[N] \times \text{frame}[F]}, \quad [0,1]\}$

$h_{\text{mean}} \leftarrow \{2000^{\text{spot}[K] \times \text{AOI}[N] \times \text{frame}[F]}, \quad \mathbb{R}_{>0}\}$

$h_{\text{beta}} \leftarrow \{0.001^{\text{spot}[K] \times \text{AOI}[N] \times \text{frame}[F]}, \quad \mathbb{R}_{>0}\}$

$w_{\text{mean}} \leftarrow \{1.5^{\text{spot}[K] \times \text{AOI}[N] \times \text{frame}[F]}, \quad [0.75, 2.25]\}$

$w_{\text{size}} \leftarrow \{100^{\text{spot}[K] \times \text{AOI}[N] \times \text{frame}[F]}, \quad \mathbb{R}_{>2}\}$

$x_{\text{mean}} \leftarrow \{0^{\text{spot}[K] \times \text{AOI}[N] \times \text{frame}[F]}, \quad [-(P+1)/2, (P+1)/2]\}$

$y_{\text{mean}} \leftarrow \{0^{\text{spot}[K] \times \text{AOI}[N] \times \text{frame}[F]}, \quad [-(P+1)/2, (P+1)/2]\}$

$xy_{\text{size}} \leftarrow \{200^{\text{spot}[K] \times \text{AOI}[N] \times \text{frame}[F]}, \quad \mathbb{R}_{>2}\}$

2: $g \sim \textbf{Gamma}(g_{\text{mean}}, \sqrt{g_{\text{mean}}/g_{\text{beta}}})$ ▷ camera gain

3: $\sigma^{xy} \sim \textbf{AffineBeta}(\sigma^{xy}_{\text{mean}}, \sigma^{xy}_{\text{size}}, 0, (P+1)/\sqrt{12})$ ▷ std of on-target spot position (pixels)

4: $\pi \sim \textbf{Beta}(\pi_{\text{mean}}, \pi_{\text{size}})$ ▷ average specific binding probability

5: $\lambda \sim \textbf{Gamma}(\lambda_{\text{mean}}, \sqrt{\lambda_{\text{mean}}/\lambda_{\text{beta}}})$ ▷ non-specific binding density

6: **for all** $\text{AOI}[N + N_c]$ **do**

7: 　　$\mu^b \sim \textbf{Delta}(\mu^b_{\text{mean}})$ ▷ mean background intensity

8: 　　$\sigma^b \sim \textbf{Delta}(\sigma^b_{\text{mean}})$ ▷ std of background intensity

9: 　　**for all** $\text{frame}[F]$ **do**

10: 　　　　$b \sim \textbf{Gamma}(b_{\text{mean}}, \sqrt{b_{\text{mean}}/b_{\text{beta}}})$ ▷ background intensity

11: 　　　　**for all** $\text{spot}[K]$ **do**

12: 　　　　　　$m \sim \textbf{Bernoulli}(m_{\text{prob}})$ ▷ spot presence

13: 　　　　　　**if** $m = 1$ **then**

14: 　　　　　　　　$h \sim \textbf{Gamma}(h_{\text{mean}}, \sqrt{h_{\text{mean}}/h_{\text{beta}}})$ ▷ spot intensity

15: 　　　　　　　　$w \sim \textbf{AffineBeta}(w_{\text{mean}}, w_{\text{size}}, 0.75, 2.25)$ ▷ spot width

16: 　　　　　　　　$x \sim \textbf{AffineBeta}(x_{\text{mean}}, xy_{\text{size}}, -(P+1)/2, (P+1)/2)$ ▷ x-axis center

17: 　　　　　　　　$y \sim \textbf{AffineBeta}(y_{\text{mean}}, xy_{\text{size}}, -(P+1)/2, (P+1)/2)$ ▷ y-axis center

18: 　　　　　　**else if** $m = 0$ **then**

19: 　　　　　　　　$h \sim \textbf{HalfNormal}(10000)$

20: 　　　　　　　　$w \sim \textbf{Uniform}(0.75, 2.25)$

21: 　　　　　　　　$x \sim \textbf{Uniform}(-(P+1)/2, (P+1)/2)$

22: 　　　　　　　　$y \sim \textbf{Uniform}(-(P+1)/2, (P+1)/2)$

**Figure 9.** Pseudocode representation of *cosmos* guide.

The online version of this article includes the following source data for figure 9:

**Source data 1.** Original text for *Figure 9*.

$$p(\text{specific}) \equiv p(z = 1 | D) \tag{24}$$

For simplicity in the main text and figures we suppress the conditional dependency on $D$ in $p(\theta | D)$ and $p(m | D)$ and instead write them as $p(\theta)$ and $p(m)$, respectively.

## Tapqir implementation

The model and variational inference method outlined above are implemented as a probabilistic program in the Python-based probabilistic programming language (PPL) Pyro (*Foerster et al., 2018*; *Bingham et al., 2019*; *Obermeyer et al., 2019a*). We use a variational approximation because exact

inference is not analytically tractable for a model as complex as *cosmos*. As currently implemented in Pyro, variational inference is significantly faster than Monte Carlo inference methods. In Tapqir, the objective that is being optimized is the evidence lower bound (ELBO) estimator that provides unbiased gradient estimates upon differentiation. At each iteration of inference procedure we choose a random subset of AOIs and frames (mini-batch), compute a differentiable ELBO estimate based on this mini-batch and update the variational parameters via automatic differentiation. We use PyTorch's Adam optimizer (*Kingma and Ba, 2014*) with the learning rate of $5 \times 10^{-3}$ and keep other parameters at their default values.

## Credible intervals and confidence intervals

Credible intervals were calculated from posterior distribution samples as the highest density region (HDR), the narrowest interval with probability mass 95% using the pyro.ops.stats.hpdi Pyro function. Confidence intervals were calculated from bootstrap samples as the 95% HDR.

## Data simulation

Simulated data were produced using the generative model (*Figure 8*). Each simulation has a subset of parameters ($\pi$, $\lambda$, $g$, $\sigma^{xy}$, $b$, $h$, $w$, $\delta$) set to desired values while the remaining parameters ($z$, $\theta$, $m$, $x$, $y$) and resulting noisy images ($D$) are sampled from distributions. The fixed parameter values and data set sizes for all simulations are provided in *Supplementary file 1*; *Supplementary file 2*; *Supplementary file 3*; *Supplementary file 4*; *Supplementary file 5*; *Supplementary file 6*.

For kinetic simulations (*Figure 5*, *Supplementary file 5*), $z$ was modeled using a discrete Markov process with the initial probability and the transition probability matrices:

$$p(z_{\text{frame}(0)} \mid k_{\text{on}}, k_{\text{off}}) = \textbf{Categorical}\left(\left[\frac{k_{\text{off}}}{k_{\text{on}}+k_{\text{off}}} \quad \frac{k_{\text{on}}}{k_{\text{on}}+k_{\text{off}}}\right]\right) \tag{25a}$$

$$p(z_{\text{frame}(f)} \mid z_{\text{frame}(f-1)}, k_{\text{on}}, k_{\text{off}}) = \textbf{Categorical}\left(\begin{bmatrix} 1 - k_{\text{on}} & k_{\text{on}} \\ k_{\text{off}} & 1 - k_{\text{off}} \end{bmatrix}\right) \tag{25b}$$

where $k_{\text{on}}$ and $k_{\text{off}}$ are transition probabilities that numerically approximate the pseudo-first-order binding and first-order dissociation rate constants in units of $\text{s}^{-1}$, respectively, assuming 1 s/frame. We assumed that the Markov process is at equilibrium and initialized the chain with the equilibrium probabilities.

## Posterior predictive sampling

For posterior predictive checking, sampled images ($\widetilde{D}$) were produced using Tapqir's generative model (*Figure 8*) where model parameters were sampled from the posterior distribution $p(\phi|D)$, which was approximated by the variational distribution $q(\phi)$:

$$\widetilde{D} \sim p(\widetilde{D} \mid D) = \int_{\phi} p(\widetilde{D} \mid \phi) p(\phi \mid D) d\phi$$

$$\simeq \int_{\phi} p(\widetilde{D} \mid \phi) q(\phi) d\phi \tag{26}$$

## Signal-to-noise ratio

We define SNR as:

$$\text{SNR} = \text{mean}\left(\frac{\text{signal}}{\sqrt{\sigma_{\text{offset}}^2 + \sigma_{\text{background}}^2}}\right) \tag{27}$$

where $\sigma_{\text{background}}^2 = b \cdot g$ the variance of the background intensity, $\sigma_{\text{offset}}^2$ the variance of the offset intensity, and the mean is taken over all target-specific spots. For experimental data, signal is calculated as

$$\text{signal} = \sum_{\substack{\text{pixelX} \\ \text{pixelY}}} (D - b_{\text{mean}} - \delta_{\text{mean}}) \cdot \text{weight} \tag{28}$$

where weight is

$$\text{weight} = \frac{1}{2\pi \cdot w^2} \exp\left(-\frac{(i - x - x^{\text{target}})^2 + (j - y - y^{\text{target}})^2}{2 \cdot w^2}\right) \tag{29}$$

For simulated data theoretical signal is directly calculated as:

$$\text{signal} = \sum_{\substack{\text{pixelX} \\ \text{pixelY}}} h \cdot \text{weight}^2 \tag{30}$$

## Classification accuracy statistics

As a metric of classification accuracy we use three commonly used statistics – recall, precision, and Matthews Correlation Coefficient (**Matthews, 1975**)

$$\text{Recall} = \frac{\text{TP}}{\text{TP} + \text{FN}} \tag{31}$$

$$\text{Precision} = \frac{\text{TP}}{\text{TP} + \text{FP}} \tag{32}$$

$$\text{MCC} = \frac{\text{TP} \cdot \text{TN} - \text{FP} \cdot \text{FN}}{\sqrt{(\text{TP} + \text{FP})(\text{TP} + \text{FN})(\text{TN} + \text{FP})(\text{TN} + \text{FN})}} \tag{33}$$

where TP is true positives, TN is true negatives, FP is false positives, and FN is false negatives.

## Kinetic and thermodynamic analysis

To estimate simple binding/dissociation kinetic parameters (**Figure 5C and D**), we sample binary time records $z$ from the inferred $p(\text{specific})$ time records for all AOIs. For a two-state hidden Markov model, the maximum-likelihood estimates of $k_{\text{on}}$ and $k_{\text{off}}$ are given by:

$$\hat{k}_{\text{on}}, \hat{k}_{\text{off}} = \underset{k_{\text{on}}, k_{\text{off}}}{\text{argmax}} \prod_{\text{AOI}} \left[ p(z_{\text{frame}(0)} \mid k_{\text{on}}, k_{\text{off}}) \prod_{f=1}^{F-1} p(z_{\text{frame}(f)} \mid z_{\text{frame}(f-1)}, k_{\text{on}}, k_{\text{off}}) \right] \tag{34}$$

Repeating this procedure 2,000 times gave the distributions of $k_{\text{on}}$ and $k_{\text{off}}$ from which we compute mean and 95% credible interval.

Similarly, to estimate mean and 95% CI of $K_{\text{eq}}$ (**Figure 5E**) we sampled $\pi$ from $q(\pi)$ and for each sampled value of $\pi$ calculated $K_{\text{eq}}$ as:

$$K_{\text{eq}} = \frac{\pi}{1 - \pi} \tag{35}$$

To calculate time-to-first binding kinetics from the Tapqir-derived $p(\text{specific})$ (**Figure 6B**, **Figure 6— figure supplement 1B**, **Figure 6—figure supplement 2B**, and **Figure 6—figure supplement 3B**), 2,000 binary time records $z$ were sampled from the $p(\text{specific})$ time record for each AOI. For each sampled time record initial absent intervals were measured and analyzed using Equation 7 in **Friedman and Gelles, 2015**, yielding distributions of $k_{\text{a}}$, $k_{\text{ns}}$, and $A_{\text{f}}$. Mean value and 95% credible intervals were calculated from these distributions. Initial absent intervals from 'spot-picker' analysis (**Figure 6C**, **Figure 6—figure supplement 1C**, **Figure 6—figure supplement 2C**, and **Figure 6—figure supplement 3C**) were analyzed as described in **Friedman and Gelles, 2015**, except that on-target and off-target data were here analyzed jointly instead of being analyzed sequentially (**Friedman and Gelles, 2015**). Note that the $k_{\text{ns}}$ values determined using the two methods are not directly comparable for several reasons, including that the non-specific binding frequencies are effectively measured over different areas. For Tapqir, the target area is approximately $\pi \left(\sigma^{xy}\right)^2$ (which is between 0.3 and 0.8 pixels$^2$ in the different experimental data sets) and for spot-picker the area is subjectively chosen as $\pi \cdot 1.5^2 = 7$ pixels$^2$.

## Acknowledgements

This work was supported by grants R01GM121384 and R01GM081648 from the National Institute of General Medical Sciences, NIH. We thank Alex Okonechnikov for his work on an earlier version of this project. We thank Jane Kondev, Timothy M Lohman, and Timothy O Street for helpful comments on the manuscript.

## Additional information

### Funding

| Funder | Grant reference number | Author |
| --- | --- | --- |
| National Institute of General Medical Sciences | R01GM121384 | Jeff Gelles<br>Douglas L Theobald |
| National Institute of General Medical Sciences | R01GM081648 | Jeff Gelles |

The funders had no role in study design, data collection and interpretation, or the decision to submit the work for publication.

### Author contributions

Yerdos A Ordabayev, Conceptualization, Investigation, Methodology, Software, Supervision, Validation, Visualization, Writing - original draft, Writing - review and editing; Larry J Friedman, Conceptualization, Data curation, Investigation, Resources, Software, Writing - review and editing; Jeff Gelles, Conceptualization, Funding acquisition, Supervision, Writing - review and editing; Douglas L Theobald, Conceptualization, Funding acquisition, Methodology, Supervision, Writing - review and editing

### Author ORCIDs

Yerdos A Ordabayev http://orcid.org/0000-0002-1493-9364
Larry J Friedman http://orcid.org/0000-0003-4946-8731
Jeff Gelles http://orcid.org/0000-0001-7910-3421
Douglas L Theobald http://orcid.org/0000-0002-2695-8343

### Decision letter and Author response

Decision letter https://doi.org/10.7554/eLife.73860.sa1
Author response https://doi.org/10.7554/eLife.73860.sa2

## Additional files

### Supplementary files

• Supplementary file 1. Varying non-specific binding rate simulation parameters and corresponding fit values.

• Supplementary file 2. Randomized simulation parameters and corresponding fit values.

• Supplementary file 3. Varying intensity (SNR) simulation parameters and corresponding fit values.

• Supplementary file 4. No target-specific binding and varying non-specific binding rate simulation parameters and corresponding fit values.

• Supplementary file 5. Kinetic simulation parameters and corresponding fit values.

• Supplementary file 6. Varying proximity simulation parameters and corresponding fit values.

• Transparent reporting form

### Data availability

All data generated or analyzed for this study will be available at https://github.com/ordabayevy/tapqir-overleaf. That repository also includes all figures, figure supplements, and the scripts and data used to generate them. It also contains the supplemental data files and preprint manuscript text.

The following dataset was generated:

| Author(s) | Year | Dataset title | Dataset URL | Database and Identifier |
|---|---|---|---|---|
| Ordabayev YA, Friedman LJ, Gelles J, Theobald DL | 2022 | Simulated and experimental data used for "Bayesian machine learning analysis of single-molecule fluorescence colocalization images" | https://github.com/ordabayevy/tapqir-overleaf | Github, ordabayevy/tapqir-overleaf |

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
