## [Editor Report]

Using a Bayesian machine learning approach, the authors of this paper have developed a tool for the analysis of single-molecule fluorescence colocalization microscopy images. The authors develop the algorithm, generate an associated software program, and then benchmark the algorithm and software using both simulated and experimental data. The results provide an important, validated tool for use by the single-molecule fluorescence microscopy community.

---

## [Decision Letter]

**Decision letter after peer review:**

Thank you for submitting your article "Bayesian machine learning analysis of single-molecule fluorescence colocalization images" for consideration by *eLife*. Your article has been reviewed by 3 peer reviewers, one of whom is a member of our Board of Reviewing Editors, and the evaluation has been overseen by José Faraldo-Gómez as the Senior Editor. The following individual involved in review of your submission has agreed to reveal their identity: Colin Kinz-Thompson (Reviewer #2).

Essential revisions:

1) The γ-distributed noise model used in Tapqir captures quite a lot of physics and, given the analyses in Figures 3-6, clearly works, but might be limited to certain types of cameras used in the fluorescence microscopy (e.g., EMCCDs). For instance, sCMOS cameras have pixel-dependent amplification and noise profiles, rather than a single gain parameter, and are sometimes approximately modeled as normal distributions with both mean and variance having intensity-dependent and intensity-independent contributions that are different for each pixel on the camera. The authors should therefore address the question of whether Tapqir can also be used with data collected from different cameras and, specifically, sCMOS cameras.

2) Little information is included about the role of AOI selection. The authors should address the question of how precisely do AOI positions need to be determined and how accurate must the mapping be. Moreover, it appears that the authors use a very large AOI (14x14 pixels, page 18). The authors should therefore address what the dependency is of the analysis on the AOI size and the relative sizes of the AOI and diffraction-limited spots. The authors should also address the question of whether Tapqir can only be used on well-separated, non-overlapping AOIs.

3) The authors should test how the strength of the prior on the location of a specific binder affects the performance of Tapqir. Specifically, it would be very informative to know how the performance of Tapqir degrades as this prior is weakened. In other words, the authors should determine how weak this prior be made before the performance of Tapquir is compromised.

4) The premise of Tapqir assumes that there is no significant "dark" population of tethered molecules on the slide. While this may hold true for the commercially synthesized DNA oligos used in the example data in Table 1, many single-molecule experiments involve a large population of dark, tethered molecules due to incomplete labeling (50% is not uncommon). In these cases, the appropriate control for non-specific binding is a separate experiment in which no molecules or control molecules are tethered to the slide surface. The authors should address the question of whether Tapqir be used in these instances.

5) The head-to-head comparison with "spot-picker" is a bit unconvincing--how commonly is spot-picker being used outside the Gelles lab? A head-to-head comparison with the previously published methods developed by Grunwald would be much more revealing--particularly concerning the utility of having event detection probabilities and the time each analysis method takes to run on the same GPU.

6) The authors offer a new approach to analyzing single-molecule fluorescence colocalization data, yet they don't leverage the full strength of priors in stringing together experiments. Specifically, in their analyses, they mostly end up in a place where their software is recapitulating old analyses and not really leveraging the probabilities they have. The authors, therefore, need to make a clearer case for using the calculated event probabilities. In addition, if they made a subjective decision about what probability cut off to use to identify events for inclusion in the kinetic modeling then that seems to go counter to the objective-based analysis of Tapqir. Can the kinetic modeling include all of the events, weighted by their probabilities?

7) The authors should also respond to the additional concerns raised by the individual reviewers, included below.

*Reviewer #1 (Public Review):*

"Bayesian machine learning analysis of single-molecule fluorescence colocalization images" by Ordabayev, et al. reports the development, benchmarking, and testing of a Bayesian machine learning-based method, which the authors name Tapqir, for analyzing single-molecule fluorescence colocalization data. Unlike currently available, more conventional analysis methods, Tapqir attempts to holistically model the microscopy images that are recorded during a colocalization experiment. Tapir uses a physics-based, global model with parameters describing all of the features of the experiment that are expected to contribute to the recorded microscopy images, including shot noise of the spots and background, camera noise, size and shape of the spots, and specific- and non-specific binders. Based on benchmarking on simulated data with widely varying properties (e.g., signal-to-noise; amounts, rates, and locations of specific and non-specific binders; etc.), Tapqir generally does as well and, in some cases, better than currently existing methods. The authors also test Tapqir on real microscopy images with similarly varying properties from studies that have been previously published by their research group and demonstrate that their Tapqir-based analysis is able to faithfully reproduce the previously published results, which were obtained using the more conventional analysis methods available at the time the data were originally published. This is a well-designed and executed study, Tapqir represents a conceptual and practical advance in the analysis of single-molecule fluorescence colocalization experiments, and its performance has been comprehensively and rigorously benchmarked on simulated data and tested on real data. The conclusions of this study are well supported by the data, but some of the limitations of the method need to be clarified and discussed in more depth, as outlined below.

1. Given that the AOI is centered at the target molecule and there is a strong prior for the binder also being located at the center of the AOI, the performance of Tapqir is dependent on several variables of the microscopy/optical system (e.g., the microscope point-spread function, magnification, accurate alignment of target and binder imaging channels, accurate drift correction, etc.). Although this caveat is mentioned and some of these factors are listed in the main text of the manuscript, the authors could have expanded this discussion in order to clarify the extent to which the performance of Tapqir depends on these factors.

2. The Tapqir model has many parameters, each with its own prior. The majority of these priors are designed to be uninformative and/or weak and the only very strong prior is the probability that a specific binder is located at or very near the center of the AOI. The authors could have tested and commented on how the strength of the prior on the location of a specific binder affects the performance of Tapqir.

3. Given the priors and variational parameters they report, the authors show that Tapqir performs robustly and seems to require no experiment-to-experiment optimization. This is expected to be the case for the simulated data, since they were simulated using the same model that Tapqir uses to perform the analysis. With regard to the real data, however, it is quite likely that this is due to the fact that the analyzed data all come from the same laboratory and, therefore, likely the same microscope(s). It would have therefore been very useful if the authors would have listed and discussed which microscope settings, experimental conditions, and/or other considerations, beyond those described in point 1 above, would result in a need for re-optimization of the priors and/or variational parameters.

4. Based on analysis of the simulated data shown in Figure 5, where the ground truth is known, the use of Tapqir to infer kinetics is less accurate that the use of Tapqir to infer equilibrium binding constants. The authors do a great job of discussing possible reasons for this. In the case of the real data analyzed in Figure 6 and in Figure 6 —figure supplements 1 and 2, the kinetic results obtained using Tapqir have different means and generally larger error bars than those obtained using Spot-Picker. To more comprehensively assess the performance of Tapqir versus Spot-Picker, the authors could have used the association and dissociation rates to calculate the corresponding equilibrium binding constants and then compared these kinetically calculated equilibrium binding constants to the population-calculated equilibrium binding constants that the authors calculate and report in the bottom plot in Panel D of Figure 6 and Figure 6 —figure supplements 1 and 2. This would provide some information on the accuracy of the kinetics in that the closer the kinetically and population-calculated equilibrium binding constants are to each other, the more accurately the kinetics have been estimated. Performing this type of analysis for the kinetics obtained using Tapqir and Spot-Picker would have allowed a more comprehensive comparison of the two methods.

*Reviewer #1 (Recommendations for the authors):*

This is a well-designed and executed study, Tapqir represents a conceptual and practical advance in the analysis of single-molecule fluorescence colocalization experiments, and its performance has been comprehensively and rigorously benchmarked on simulated data and tested on real data. Moreover, the conclusions of this study are well supported by the data. Given all of this, I would recommend publication of this study as a Tools and Resources article in *eLife*, assuming that the authors address the weaknesses I identified in my Public Review as well as the following extensions of those weaknesses.

1. I think the authors should to expand the discussion of how the performance of Tapqir depends on the microscope point-spread function, magnification, accurate alignment of target and binder imaging channels, accurate drift correction, etc. Specifically, if possible, they should describe how a 1-2 pixel offset in the x- or y dimensions between the target and binder imaging channels arising from differences in any of these parameters would affect the performance of Tapqir. This is especially important given the strength of the prior the authors have assigned to the location of the specific binder at the center of the AOI.

2. The authors should list and discuss what microscope settings, experimental conditions, and/or other considerations, beyond the microscope/optical described in point 1 above, would result in a need for re-optimization of the priors and/or variational parameters. For example, in Lines 509-510, the authors state that most microscopes used for colocalization experiments are set up such that diffraction-limited spots occupy 1-2 pixels in the x- and y dimensions on the camera detector. If a microscope is instead set up to spread the spot over 3, 4, or more pixels in each dimension, are there any priors or variational parameters that should be re-optimized? Are there any other such considerations?

4. The authors should comment on whether the kinetic parameters obtained by Tapqir and reported in Figure 6 and in Figure 6 —figure supplements 1 and 2 are actually more accurate and/or precise than those obtained by Spot-Picker. For example, if the association and dissociation rates were used to calculate the equilibrium binding constants, how would these kinetically calculated equilibrium binding constants compare to the population-calculated equilibrium constant that the authors calculate and report in Figure 6 and in Figure 6 —figure supplements 1 and 2? Are the kinetically calculated and population-calculated equilibrium binding constants in closer agreement for the Tapqir-analyzed data versus the Spot-Picker-analyzed data? If one is better than the other, why do the authors think that is?

*Reviewer 2 (Public Review):*

The work by Ordabayev et al. details a Bayesian inference-based data analysis method for colocalization single molecule spectroscopy (CoSMoS) experiments used to investigate biochemical and biophysical mechanisms. By using this probabilistic framework, their method is able to quantify the colocalization probabilities for individual molecules while accounting for the uncertainty in individual binding events, and accounting for camera and optical noise and even non-specific binding. The software implementation of this method, called Tapqir, uses a Python-based probabilistic programming language (PPL) called pyro to automate and speed-up the optimization of a variational Bayes approximation to the posterior probability distribution. Overall, Tapqir is a powerful new way to analyze CoSMoS data.

Tapqir works by analyzing small regions (14x14 pixels) of fluorescence microscopy images surrounding previously identified areas of interest (AOI). The collection of images of these AOIs through time are then analyzed collectively using a probabilistic model that accounts for each time frame of each AOI and is able to determine whether up to K "binders" (K=2 here) are present and which of them is specifically bound. This approach of directly modeling the contents of the image data is relatively novel, and few other examples exist. The details of the probabilistic model used incorporate an impressive amount of physical insight (e.g., camera gain) without overparameterization.

The γ-distributed noise model used in Tapqir captures quite a lot of physics and, given the analyses in Figures 3-6, clearly works, but might be limited to certain types of cameras used in the fluorescence microscopy (e.g., EMCCDs). For instance, sCMOS cameras have pixel-dependent amplification and noise profiles, rather than a single gain parameter, and are sometimes approximately modeled as normal distributions with both mean and variance having an intensity-dependent and independent contribution that is different for each pixel on the camera. It is unclear how Tapqir performs on different cameras.

The variational Bayes solution used by Tapqir provides many computational benefits, such as numerical tractability using pyro and speed. It is possible that the exact posterior, e.g., as obtained using a Markov chain Monte Carlo method, would be insignificantly different with the amount of data typical for CoSMoS experiments; however, this difference is not explored in the current work.

The intrinsic use of prior probability distributions in any Bayesian inference algorithm is extremely powerful, and in Tapqir offers the opportunity to "chain together" subsequent analyses by using the marginalized posteriors from one experiment as the basis for the priors for subsequent experiments (e.g., in \σ^{xy}) for extremely high accuracy inference. While the manuscript discusses setting and leveraging the power of priors, it does not explore the power of such "chaining" and the positive effects upon accuracy.

A significant number of CoSMoS experiments use multiple, distinct color fluorophores to probe the colocalization of different species to the target. The current work focuses only upon analyzing data with a single color-channel. Extensions to multiple independent wavelengths are computationally trivial, given the automated variational inference ability of PPLs such as pyro, and would increase the impact of the work in the field.

Tapqir analysis provides time series of the probability of a specific binding event, p(specific), for each target analyzed (c.f., Figure 5B), and kinetic parameters are extracted from these time series using secondary analyses that are distinct from Tapqir itself.

The method reported here is well designed, sound, and its utility is well supported by the analyses of simulated and experimental data sets reported here. Tapqir is a cutting-edge image analysis approach, and its proper treatment of the uncertainty inherent to CoSMoS experiments will certainly make an impact upon the analysis of CoSMoS data. However, many of the (necessary) assumptions about the data (e.g., fluorescence microscopy) and desired information (e.g., off-target vs on-target binding) are quite specific to CoSMoS experiments and therefore limit the direct applicability of Tapqir for the analysis of other single-molecule microscopy techniques. With that in mind, the direct Bayesian inference-based analysis of image data, as opposed to integrated time series, as demonstrated here is very powerful, and may encourage and inspire related methods to be developed.

*Reviewer #2 (Recommendations for the authors):*

– Some of the language in the introduction is a little imprecise (e.g., "binders", "green RNA", "blue DNA spot", "integrating binder fluorescence", "real fluorescent spots"), and could be more explicit to improve clarity.

– Line 63: The concentration barrier could be described more in depth for the *eLife* readership.

– Line 74-76: Additional description of these effects, perhaps mathematically or through other citations, would help the readers understand the fundamental differences between analyzing image data and intensity data.

– Line 82-83: Describing how and/or the magnitude of the failure that not accounting for spot confidence creates would be useful for the reader to understand the requirement for Tapqir

– Line 84-86: The method described doesn't get a name, but the software does get a name. I think giving the method a descriptive name (e.g., an acronym) would help clarify the discussion and distinction between the approach of probabilistic modeling of the data and using pyro and the chosen priors etc. to do so.

– More referencing of Bayesian image analysis methods for microscopy data, at least in the introduction, (e.g., Bayesian Analysis of Blinking and Bleaching (B3), maybe some super-resolution methods, etc.) would help create the appropriate context for Tapqir.

– A discussion of the benefits of variational as opposed to exact inference is missing and would be useful for the reader.

– Line ~139: It is unclear if the image models or PSFs are integrated over pixel boundaries (i.e., as in Smith et al., "Fast, single-molecule localization that achieves theoretically minimum uncertainty" DOI: 10.1038/nmeth.1449). If not, what effect does this have on the modeling?

– Line 155-161: A discussion of EMCCD versus sCMOS noise differences, or even which one is more applicable to Tapqir, would be helpful here.

– Line 181: It is unclear what the "hierarchichal Bayesian analysis" refers to. I could not find an explanation in the Methods.

– Figure 3: What is the criteria for not having {x,y,h} included on the plot (e.g., at t=101)? I could not find it. Maybe p(m=0)>.5?

– Figure 3: This figure should also include w along with x,y, and h. Is it relatively constant? Does it vary quite a bit?

– Figure 3-supplement 1 A: It was unclear to me why at frame 103, the spot was detected as spot 2 and not spot 1 with equal probability. Isn't there a degeneracy between the two spots? Is this broken by \theta? Regardless of the answer, perhaps a more in-depth discussion of this point would be useful.

– Figure 3-supplement 3 D: How does this plot compare to the theoretical minimum uncertainty of localizing a single molecule (i.e., the Cramer-Rao lower bound) at these photon fluxes? Shouldn't it bottom out at some point?

– Line 214: "… rich enough to accurately capture …" is a very nice way to convey the utility of the model. I think you should use it more often.

– Figure 5C-E: the rate constants are systematically overestimated -- even at the slowest rates. Why? Might these rates constants actually transition probabilities? I did not see a k=-ln(1-P)/dt equation in the Methods section.

– Line 353: Generally, Tapqir is only quantitatively compared to spot-picker, when there is also the Carlas et al., method that could be used for comparison.

– Figure 6B and supplements: Why were no off target controls ever analyzed to be included in the plot B as the yellow curve in C. If nothing else, it would be very useful to show the Tapqir is very accurate.

– Table 1: The computation times are reasonable for a high quality analysis, but are done on a very fast desktop computer (Threadripper with a 2080). It would be useful to show the performance on a less powerful computer as well (e.g., a low-powered laptop) for a least one dataset or perhaps a partial dataset. That way, potential users can judge whether they need to seek out better computational resources before trying Tapqir.

*Reviewer #3 (Public Review):*

In this manuscript, the authors seek to improve the reproducibility and eliminate sources of bias in the analysis of single molecule colocalization fluorescence data. These types of data (i.e., CoSMoS data) have been obtained from a number of diverse biological systems and represent unique challenges for data analysis in comparison with smFRET. A key source of bias is what constitutes a binding event and if those events are colocalized or not with a surface-tethered molecule of interest. To solve these issues, the authors propose a Bayesian-based method in which each image is analyzed individually and locally around areas of interest (AOIs) identified from the surface tethered molecules. A strength of the research is that the approach eliminates many sources of bias (i.e., thresholding) in analysis, models realistic image features (noise), can be automated and carried out by novice users "hands-free", and returns a probability score for each event. The performance of the method is superb under a number of conditions and with varying levels of signal-to-noise. The analysis on a GPU is fairly quick-overnight-in comparison with by-hand analysis of the traces which can take days or longer. Tapqir has the potential to be the go-to software package for analysis of single molecule colocalization data.

The weaknesses of this work involve concerns about the approach and its usefulness to the single-molecule community at large as wells as a lack of information about how users implement and use the Tapqir software. For the first item, there are a number of common scenarios encountered in colocalization analysis that may exclude use of Tapqir including use of CMOS rather than EM-CCD cameras, significant numbers of tethered molecules on the surface that are dark/non-fluorescent, a high density/overlapping of AOIs, and cases where event intensity information is critical (i.e., FRET detection or sequential binding and simultaneous occupancy of multiple fluorescent molecules at the same AOI). In its current form, the use of Tapqir may be limited to only certain scenarios with data acquired by certain types of instruments.

Second, for adoption by non-expert users information is missing in the main text about practical aspects of using the Tapqir software including a description of inputs/outputs, the GUI (I believe Taqpir runs at the command line but the output is in a GUI), and if Tapqir integrates the kinetic modeling or not. Given that a competing approach has already been published by the Grunwald lab, it would be useful to compare these methods directly in both their accuracy, usefulness of the outputs, and calculation times. Along these lines, the utility of calculating event probability statistics (Figure 6A) is not well fleshed-out. This is a key distinguishing feature between Tapqir and methods previously published by Grunwald et al. In the case of Tapqir, the probability outputs are not used to their fullest in the determination of kinetic parameters. Rather a subjective probability threshold is chosen for what events to include. This may introduce bias and degrade the objective Tapqir pipeline used to identify these same events.

Finally, the manuscript could be improved by clearly distinguishing between the fundamental approach of Bayesian image analysis from the Tapqir software that would be used to carry this out. A section devoted to describing the Tapqir interface and the inputs/outputs would be valuable. In the manuscript's current form, the lack of information on the interface along with the potential requirement for a GPU and need for the use of a relatively new programming language (Pyro) may hamper adoption and interest in colocalization methods by general audiences.

*Reviewer #3 (Recommendations for the authors):*

1. It is unclear if intensity information is used by Tapqir or if it can be used. This can be useful for including more priors about the experiment (i.e., "real" events would be above a certain threshold due to FRET or presence of multiple fluorophores) or for using Tapqir to analyze experiments in which multiple fluorophore-labeled molecules bind simultaneously and sequentially to the same AOI. As presented, it would seem that Tapqir is "blind" to these types of multiple binding events.

2. A concern for adoption of Tapqir and appreciation of this work by general audiences involves the presentation of the method and software. I think that these should be disentangled from one another and that Tapqir should only be used to refer to the software used to carry out this approach. The manuscript, and the colocalization field, may be better served if a section were included that explicitly describes how to use Tapqir to implement this analysis including the necessary inputs, hardware (how much time would this take if a GPU isn't used?), and outputs/GUI. Ultimately, Tapqir needs to be user-friendly to be adopted and the requirement for a GPU and the Pyro programming language may be significant barriers. A potential model for the authors to consider is the *eLife* paper describing cisTEM software (https://elifesciences.org/articles/35383) that efficiently describes both the process, benchmarking, and software/user experience.

3. With respect to inputs, the need for use of imscroll to identify AOIs, drift correct, and carry out mapping should be clarified. Is imscroll output essential for Tapqir input?

---

## [Author Response]

Essential revisions:1) The γ-distributed noise model used in Tapqir captures quite a lot of physics and, given the analyses in Figures 3-6, clearly works, but might be limited to certain types of cameras used in the fluorescence microscopy (e.g., EMCCDs). For instance, sCMOS cameras have pixel-dependent amplification and noise profiles, rather than a single gain parameter, and are sometimes approximately modeled as normal distributions with both mean and variance having intensity-dependent and intensity-independent contributions that are different for each pixel on the camera. The authors should therefore address the question of whether Tapqir can also be used with data collected from different cameras and, specifically, sCMOS cameras.

We agree that this point should be addressed. We have expanded the discussion of the Image likelihood component of our model (lines 481 – 502) to emphasize that (1) all data sets we analyze are experimental or simulated EMCCD images, (2) sCMOS images have the different noise characteristics alluded to by the reviewers, and (3) optimal sCMOS image analysis might require a modified model, possibly including the ability to use per-pixel calibration data as a prior as was done in super-resolution work (now cited) that uses sCMOS data.

sCMOS cameras have in recent years become very popular for some kinds of single-molecule imaging (e.g., PALM/STORM or live-cell single-particle tracking). However, for the low-background/low-signal in vitro single-molecule TIRF that is our target application for the approach described in the manuscript, EMCCD is still preferable over sCMOS for many, but not all, imaging conditions (see https://andor.oxinst.com/learning/view/article/what-is-the-best-detector-for-single-molecule-studies). Thus, we think there will be plenty of interest in the approach we describe in the manuscript even if the program functions better with EMCCD than with sCMOS images.

Going forward to develop and test an sCMOS-targeted version of the model, as we have done for EMCCD, will require revised model and code, but will also necessitate accurately simulating sCMOS CoSMoS images, obtaining experimental sCMOS CoSMoS images reflecting a broad range of realistic experimental conditions, and using the simulated and experimental images to test the new model. These may well be useful things to do in the future but would be a considerable step beyond the scope of the present manuscript.

2) Little information is included about the role of AOI selection. The authors should address the question of how precisely do AOI positions need to be determined and how accurate must the mapping be. Moreover, it appears that the authors use a very large AOI (14x14 pixels, page 18). The authors should therefore address what the dependency is of the analysis on the AOI size and the relative sizes of the AOI and diffraction-limited spots. The authors should also address the question of whether Tapqir can only be used on well-separated, non-overlapping AOIs.

To address these questions, we added new data to the manuscript in Table 2, Table 5, and Figure 3 —figure supplement 4. The question about mapping accuracy is now discussed in the Methods (lines 565 – 574):

“Tests on data simulated with increasing proximity parameter values σ^xy^ (true) (i.e., with decreasing precision of spatial mapping between the binder and target image channels) confirm that the *cosmos* model accurately learns σ^xy^ (fit) from the data (Figure3—figure supplement 3D; Table 5). This was the case even if we substituted a less-informative σ^xy^ prior (Uniform vs. Exponential; Table 5).

The CoSMoS technique is premised on colocalization of the binder spots with the known location of the target molecule. Consequently, for any analysis method, classification accuracy declines when the images in the target and binder channels are less accurately mapped. For the Tapqir *cosmos* model, low mapping precision has little effect on classification accuracy at typical non-specific binding densities (λ = 0.15; see MCC values in Table 5).”

We also added to the Results section (lines 302 -308) the following discussion of the effect of AOI size:

“Since target-nonspecific spots are built into the *cosmos* model, there is no need to choose excessively small AOIs in an attempt to exclude non-specific spots from analysis. We found that reducing AOI size (from 14 x 14 to 6 x 6 pixels) did not appreciably affect analysis accuracy on simulated data (Table 2). In analysis of experimental data, smaller AOI sizes caused occasional changes in calculated p(specific) values reflecting apparent missed detection of a few spots (Figure 3—figure supplement 4). Out of caution, we therefore used 14 x 14 pixel AOIs routinely, even though the larger AOIs somewhat reduced computation speed (Table 2 and Figure 3—figure supplement 4).”

The method does not require non-overlapping AOIs – we used partially overlapping AOIs in the experimental data analyzed in the manuscript. Even though our analysis used larger AOI sizes (and hence, more overlap) than the spot-picker method, there was good agreement in the results, indicating that overlap does not cause any undue problems.

3) The authors should test how the strength of the prior on the location of a specific binder affects the performance of Tapqir. Specifically, it would be very informative to know how the performance of Tapqir degrades as this prior is weakened. In other words, the authors should determine how weak this prior be made before the performance of Tapquir is compromised.

In our model, the position of a target-specific spot relative to the target position has a prior distribution illustrated as the green curve in Figure 2—figure supplement 2. Importantly, the peak in this distribution does not have an a priori set width. Instead, the width of the peak is a model hyperparameter, σ^xy^, that is learned from the image data set without user intervention. To make sure that this point is understood, we expanded and clarified the relevant Methods section (the four paragraphs starting at line 549) and modified the legend of Figure 2—figure supplement 2.

To address the reviewers’ specific question, we constructed simulated data sets with different mapping precision values and analyzed them; the results are presented in the (new) Table 5 and discussed (lines 570 – 574):

“The CoSMoS technique is premised on colocalization of the binder spots with the known location of the target molecule. Consequently, for any analysis method, classification accuracy declines when the images in the target and binder channels are less accurately mapped. For the Tapqir *cosmos* model, low mapping precision has little effect on classification accuracy at typical non-specific binding densities (λ = 0.15; see MCC values in Table 5).”

4) The premise of Tapqir assumes that there is no significant "dark" population of tethered molecules on the slide. While this may hold true for the commercially synthesized DNA oligos used in the example data in Table 1, many single-molecule experiments involve a large population of dark, tethered molecules due to incomplete labeling (50% is not uncommon). In these cases, the appropriate control for non-specific binding is a separate experiment in which no molecules or control molecules are tethered to the slide surface. The authors should address the question of whether Tapqir be used in these instances.

The reviewers suggest a “no target molecules in sample” (NTIS) control instead of the “no fluorescent target molecules in control AOIs” (NFTICA) design that we illustrate in Figure 1. Both types can be used as a Tapqir control dataset without any modification of the program or model. We have edited the Figure 1 caption to explain that either type is acceptable. The reviewers are correct that, all else being equal, NTIS may be better if the target molecules are incompletely labeled. However, in practice experimenters usually know the fraction of molecules that are labeled and reduce the fluorescent target molecule surface density to hold the fraction of spots with two or more coincident target molecules (fluorescent or not) below a chosen threshold (typically 1 % or less), negating the possible advantage of NTIS (but at the expense of collecting less data per sample). On the other hand, NFTICA has the practical advantage that it is a control internal to the sample and is thus immune to problems caused by temporal or sample-to-sample variability (e.g., of surface properties).

5) The head-to-head comparison with "spot-picker" is a bit unconvincing--how commonly is spot-picker being used outside the Gelles lab? A head-to-head comparison with the previously published methods developed by Grunwald would be much more revealing--particularly concerning the utility of having event detection probabilities and the time each analysis method takes to run on the same GPU.

Both the spot-picker binary classification method and the Grunwald binary classification method have been used outside of the lab in which they originated. It’s difficult to gauge exact usage of freely available software, but PubMed says that the spot-picker paper [Friedman and Gelles *Methods* 2015] has been cited 38 times, 26 of which by papers that do not have Gelles as a co-author, while the Grunwald paper [Smith, … Zamore and Grunwald *Nat. Commun.* 2019] (which in fairness was published much more recently) has been cited 7 times, 6 of which do not have Zamore as co-author.

The reviewers do not explain why comparing with the Grunwald method would be preferable to comparing with spot-picker. To be sure there is no misunderstanding, the following are the same for the two methods and therefore are **not** reasons to prefer one or the other of these methods for the comparison in Figure 6 (see also Discussion lines 392-415):

1. Like Tapqir, both spot-picker and Grunwald methods analyze 2-D images, not integrated intensities.

2. Unlike Tapqir, neither spot-picker nor Grunwald is fully objective; both require subjective selection of classification thresholds by an expert analyst in order to tune the algorithm performance for analysis of a particular dataset.

3. Neither spot-picker nor Grunwald is a Bayesian method. “Bayesian” in the Grunwald paper title refers to a separate analytical method (described in the same paper) for evaluating the number of binder molecules colocalized with a target spot; this method is not relevant to a comparison with the model presented in the manuscript.

4. Unlike Tapqir, neither spot-picker nor Grunwald estimate classification probabilities. Instead, they simply assign binary spot/no-spot classifications that do not convey to downstream analyses the extent of uncertainty in each classification.

5. Neither spot-picker nor Grunwald has been validated previously using simulated image data. Consequently, the validity of image classification has not been established for either.

The comparison of Figure 6 and supplements (lines 365-375) does not claim to and is not intended to show that Tapqir is better than spot-picker for real experimental data; we cannot make such a claim for any method because we do not know the true kinetic process that generated our data. Instead, our comparison uses experimental data sets with a broad range of characteristics (Table 1) to show that Tapqir yields similar association rate constants to those produced by spot-picker even though the former is objective and automatic while the latter requires subjective tuning by an expert analyst. Our choice to use spot-picker over Grunwald for this comparison was dictated by the fact that among the co-authors we have such an expert in the use of spot-picker, whereas we lack comparable expertise with Grunwald. We have no doubt that Grunwald would also produce results similar to the other methods in the hands of an expert user who is able to subjectively adjust classification parameters.

6) The authors offer a new approach to analyzing single-molecule fluorescence colocalization data, yet they don't leverage the full strength of priors in stringing together experiments. Specifically, in their analyses, they mostly end up in a place where their software is recapitulating old analyses and not really leveraging the probabilities they have. The authors, therefore, need to make a clearer case for using the calculated event probabilities. In addition, if they made a subjective decision about what probability cut off to use to identify events for inclusion in the kinetic modeling then that seems to go counter to the objective-based analysis of Tapqir. Can the kinetic modeling include all of the events, weighted by their probabilities?

This comment seems to be premised on a misunderstanding of our manuscript. While we used a cut-off of *p*(specific) > 0.5 for the binary classification statistics (e.g., Figure 4D and G), all of the kinetic analyses in the manuscript use the full time series of calculated probability values with no arbitrary cut-off. This was accomplished using the posterior sampling method that is illustrated in Figure 5B and described in lines 318-324. We have revised the latter to clarify this point.

We address the specific point about use of priors in stinging together experiments in a response to Reviewer #2 (Public Review) below.

7) The authors should also respond to the additional concerns raised by the individual reviewers, included below.

Please see responses below.

Reviewer #1 (Public Review):"Bayesian machine learning analysis of single-molecule fluorescence colocalization images" by Ordabayev, et al. reports the development, benchmarking, and testing of a Bayesian machine learning-based method, which the authors name Tapqir, for analyzing single-molecule fluorescence colocalization data. Unlike currently available, more conventional analysis methods, Tapqir attempts to holistically model the microscopy images that are recorded during a colocalization experiment. Tapir uses a physics-based, global model with parameters describing all of the features of the experiment that are expected to contribute to the recorded microscopy images, including shot noise of the spots and background, camera noise, size and shape of the spots, and specific- and non-specific binders. Based on benchmarking on simulated data with widely varying properties (e.g., signal-to-noise; amounts, rates, and locations of specific and non-specific binders; etc.), Tapqir generally does as well and, in some cases, better than currently existing methods. The authors also test Tapqir on real microscopy images with similarly varying properties from studies that have been previously published by their research group and demonstrate that their Tapqir-based analysis is able to faithfully reproduce the previously published results, which were obtained using the more conventional analysis methods available at the time the data were originally published. This is a well-designed and executed study, Tapqir represents a conceptual and practical advance in the analysis of single-molecule fluorescence colocalization experiments, and its performance has been comprehensively and rigorously benchmarked on simulated data and tested on real data. The conclusions of this study are well supported by the data, but some of the limitations of the method need to be clarified and discussed in more depth, as outlined below.1. Given that the AOI is centered at the target molecule and there is a strong prior for the binder also being located at the center of the AOI, the performance of Tapqir is dependent on several variables of the microscopy/optical system (e.g., the microscope point-spread function, magnification, accurate alignment of target and binder imaging channels, accurate drift correction, etc.). Although this caveat is mentioned and some of these factors are listed in the main text of the manuscript, the authors could have expanded this discussion in order to clarify the extent to which the performance of Tapqir depends on these factors.

The revised manuscript includes new data on and expanded discussion of this point. Please see our response to Essential Revision 2.

2. The Tapqir model has many parameters, each with its own prior. The majority of these priors are designed to be uninformative and/or weak and the only very strong prior is the probability that a specific binder is located at or very near the center of the AOI. The authors could have tested and commented on how the strength of the prior on the location of a specific binder affects the performance of Tapqir.

The revised manuscript includes new data on and expanded discussion of this point. Please see response to Essential Revision 3.

3. Given the priors and variational parameters they report, the authors show that Tapqir performs robustly and seems to require no experiment-to-experiment optimization. This is expected to be the case for the simulated data, since they were simulated using the same model that Tapqir uses to perform the analysis. With regard to the real data, however, it is quite likely that this is due to the fact that the analyzed data all come from the same laboratory and, therefore, likely the same microscope(s). It would have therefore been very useful if the authors would have listed and discussed which microscope settings, experimental conditions, and/or other considerations, beyond those described in point 1 above, would result in a need for re-optimization of the priors and/or variational parameters.

We now address this point in the Materials and methods as follows (lines 587-595):

“All simulated and experimental data sets in this work were analyzed using the prior distributions and hyperparameter values given above, which are compatible with a broad range of experimental conditions (Table 1). Many of the priors are uninformative and we anticipate that these will work well with images taken on variety of microscope hardware. However, it is possible that highly atypical microscope designs (e.g., those with effective magnifications that are sub-optimal for CoSMoS) might require adjustment of some fixed hyperparameters and distributions (those in Eqs. 6a, 6b, 11, 12, 13, 15, and 16). For example, if the microscope point spread function is more than 2 pixels wide, it may be necessary to increase the range of the *w* prior in Eq. 13. The Tapqir documentation

4. Based on analysis of the simulated data shown in Figure 5, where the ground truth is known, the use of Tapqir to infer kinetics is less accurate that the use of Tapqir to infer equilibrium binding constants. The authors do a great job of discussing possible reasons for this. In the case of the real data analyzed in Figure 6 and in Figure 6 —figure supplements 1 and 2, the kinetic results obtained using Tapqir have different means and generally larger error bars than those obtained using Spot-Picker. To more comprehensively assess the performance of Tapqir versus Spot-Picker, the authors could have used the association and dissociation rates to calculate the corresponding equilibrium binding constants and then compared these kinetically calculated equilibrium binding constants to the population-calculated equilibrium binding constants that the authors calculate and report in the bottom plot in Panel D of Figure 6 and Figure 6 —figure supplements 1 and 2. This would provide some information on the accuracy of the kinetics in that the closer the kinetically and population-calculated equilibrium binding constants are to each other, the more accurately the kinetics have been estimated. Performing this type of analysis for the kinetics obtained using Tapqir and Spot-Picker would have allowed a more comprehensive comparison of the two methods.

This comment seems to reflect a misunderstanding. Figure 6 and its figure supplements do not report any dissociation kinetics or binding equilibrium constants. Instead, they report *k*_a_ (pseudo first-order association rate constant), *k*_ns_ (non-specific binding association rate constant), and *A*_f_ (the active faction, i.e., the fraction of target molecules capable of association with binder). *k*_a_ and *A*_f_ values from the two methods agree within experimental uncertainty for all four data sets analyzed. *k*_ns_ values differ, but as we point out (lines 373-375):

“We noted some differences between the two methods in the non-specific association rate constants *k*_ns_. Differences are expected because these parameters are defined differently in the different nonspecific binding models used in Tapqir and spot-picker (see Materials and methods).”

(There is additional discussion of this point in Materials and methods lines 686-690.) The reviewer is correct that the estimated uncertainties (i.e., error bars in panels D) in *k*_a_ and *A*_f_ are generally larger for Tapqir than for spot-picker. This is expected, for the reasons that we explain (lines 402-411):

“In general, previous approaches in essence assume that spot classifications are correct, and thus the uncertainties in the derived molecular properties (e.g., equilibrium constants) are systematically underestimated because the errors in spot classification, which can be large, are not accounted for. By performing a probabilistic spot classification, Tapqir enables reliable inference of molecular properties, such as thermodynamic and kinetic parameters, and allows statistically well-justified estimation of parameter uncertainties. This more inclusive error estimation likely accounts for the generally larger kinetic parameter error bars obtained from Tapqir compared to those from the existing spot-picker analysis method (Figure 6, Figure 6—figure supplement 1, Figure 6—figure supplement 2, and Figure 6—figure supplement 3). ”

Reviewer #1 (Recommendations for the authors):This is a well-designed and executed study, Tapqir represents a conceptual and practical advance in the analysis of single-molecule fluorescence colocalization experiments, and its performance has been comprehensively and rigorously benchmarked on simulated data and tested on real data. Moreover, the conclusions of this study are well supported by the data. Given all of this, I would recommend publication of this study as a Tools and Resources article in eLife, assuming that the authors address the weaknesses I identified in my Public Review as well as the following extensions of those weaknesses.This is a well-designed and executed study, Tapqir represents a conceptual and practical advance in the analysis of single-molecule fluorescence colocalization experiments, and its performance has been comprehensively and rigorously benchmarked on simulated data and tested on real data. Moreover, the conclusions of this study are well supported by the data. Given all of this, I would recommend publication of this study as a Tools and Resources article in eLife, assuming that the authors address the weaknesses I identified in my Public Review as well as the following extensions of those weaknesses.1. I think the authors should to expand the discussion of how the performance of Tapqir depends on the microscope point-spread function, magnification, accurate alignment of target and binder imaging channels, accurate drift correction, etc. Specifically, if possible, they should describe how a 1-2 pixel offset in the x- or y dimensions between the target and binder imaging channels arising from differences in any of these parameters would affect the performance of Tapqir. This is especially important given the strength of the prior the authors have assigned to the location of the specific binder at the center of the AOI.

The revised manuscript includes new data on and expanded discussion of this point. Please see response to Essential Revision 3.

2. The authors should list and discuss what microscope settings, experimental conditions, and/or other considerations, beyond the microscope/optical described in point 1 above, would result in a need for re-optimization of the priors and/or variational parameters. For example, in Lines 509-510, the authors state that most microscopes used for colocalization experiments are set up such that diffraction-limited spots occupy 1-2 pixels in the x- and y dimensions on the camera detector. If a microscope is instead set up to spread the spot over 3, 4, or more pixels in each dimension, are there any priors or variational parameters that should be re-optimized? Are there any other such considerations?

This is now addressed. Please see response to Reviewer #1 (Public Review) point 3.

4. The authors should comment on whether the kinetic parameters obtained by Tapqir and reported in Figure 6 and in Figure 6 —figure supplements 1 and 2 are actually more accurate and/or precise than those obtained by Spot-Picker. For example, if the association and dissociation rates were used to calculate the equilibrium binding constants, how would these kinetically calculated equilibrium binding constants compare to the population-calculated equilibrium constant that the authors calculate and report in Figure 6 and in Figure 6 —figure supplements 1 and 2? Are the kinetically calculated and population-calculated equilibrium binding constants in closer agreement for the Tapqir-analyzed data versus the Spot-Picker-analyzed data? If one is better than the other, why do the authors think that is?

We do not calculate dissociation rate constants or equilibrium constants in Figure 6 and its figure supplements; please see the discussion of this in response to Reviewer #1 (Public Review) point 4. As the reviewer points out, we know the “ground truth” for the simulated data of Figure 5 but not for the experimental data of Figure 6. Consequently, it is not possible assess the accuracy of analysis of the latter. Precision is already estimated; please see the discussion of error bars in response to Reviewer #1 (Public Review) point 4.

Reviewer 2 (Public Review):The work by Ordabayev et al. details a Bayesian inference-based data analysis method for colocalization single molecule spectroscopy (CoSMoS) experiments used to investigate biochemical and biophysical mechanisms. By using this probabilistic framework, their method is able to quantify the colocalization probabilities for individual molecules while accounting for the uncertainty in individual binding events, and accounting for camera and optical noise and even non-specific binding. The software implementation of this method, called Tapqir, uses a Python-based probabilistic programming language (PPL) called pyro to automate and speed-up the optimization of a variational Bayes approximation to the posterior probability distribution. Overall, Tapqir is a powerful new way to analyze CoSMoS data.Tapqir works by analyzing small regions (14x14 pixels) of fluorescence microscopy images surrounding previously identified areas of interest (AOI). The collection of images of these AOIs through time are then analyzed collectively using a probabilistic model that accounts for each time frame of each AOI and is able to determine whether up to K "binders" (K=2 here) are present and which of them is specifically bound. This approach of directly modeling the contents of the image data is relatively novel, and few other examples exist. The details of the probabilistic model used incorporate an impressive amount of physical insight (e.g., camera gain) without overparameterization.

We thank the reviewer for these positive comments.

The γ-distributed noise model used in Tapqir captures quite a lot of physics and, given the analyses in Figures 3-6, clearly works, but might be limited to certain types of cameras used in the fluorescence microscopy (e.g., EMCCDs). For instance, sCMOS cameras have pixel-dependent amplification and noise profiles, rather than a single gain parameter, and are sometimes approximately modeled as normal distributions with both mean and variance having an intensity-dependent and independent contribution that is different for each pixel on the camera. It is unclear how Tapqir performs on different cameras.

Please see our response to Essential Revisions point 1.

The variational Bayes solution used by Tapqir provides many computational benefits, such as numerical tractability using pyro and speed. It is possible that the exact posterior, e.g., as obtained using a Markov chain Monte Carlo method, would be insignificantly different with the amount of data typical for CoSMoS experiments; however, this difference is not explored in the current work.

We agree. However, since we have not done any analyses using MCMC, there is nothing in particular that we can say about it in the context of CoSMoS data analysis. Implementation of an MCMC approach using our model will be easier in the future because the pyro developers are currently working to optimize the implementations of MCMC methods in their software.

The intrinsic use of prior probability distributions in any Bayesian inference algorithm is extremely powerful, and in Tapqir offers the opportunity to "chain together" subsequent analyses by using the marginalized posteriors from one experiment as the basis for the priors for subsequent experiments (e.g., in \σ^{xy}) for extremely high accuracy inference. While the manuscript discusses setting and leveraging the power of priors, it does not explore the power of such "chaining" and the positive effects upon accuracy.

Chaining is beneficial in principle. However, in practice it will help significantly only if the uncertainty in the posterior parameter values from the non-chained analysis is larger than the experiment-toexperiment variability in the “true” parameter values. For σ^xy^ we obtain very narrow credence intervals without chaining (Table 1). In our judgement, these are unlikely to be made more accurate by using prior information from another experiment where such factors as microscope focus adjustment may be slightly different.

A significant number of CoSMoS experiments use multiple, distinct color fluorophores to probe the colocalization of different species to the target. The current work focuses only upon analyzing data with a single color-channel. Extensions to multiple independent wavelengths are computationally trivial, given the automated variational inference ability of PPLs such as pyro, and would increase the impact of the work in the field.

Our current approach can be used to analyze multi-channel data simply by analyzing each channel independently. However, we agree that there would be advantages to joint analysis of multiple wavelength channels (especially if there is crosstalk between channels) and that implementing multichannel analysis is a logical extension of our study. It is straightforward (though not trivial, in our experience) to implement such multi-wavelength models. However, testing the functioning of candidate models and validating them using simulation and experimental data would require extensive work that in our view goes beyond what is reasonable to include in the present manuscript.

Tapqir analysis provides time series of the probability of a specific binding event, p(specific), for each target analyzed (c.f., Figure 5B), and kinetic parameters are extracted from these time series using secondary analyses that are distinct from Tapqir itself.The method reported here is well designed, sound, and its utility is well supported by the analyses of simulated and experimental data sets reported here. Tapqir is a cutting-edge image analysis approach, and its proper treatment of the uncertainty inherent to CoSMoS experiments will certainly make an impact upon the analysis of CoSMoS data. However, many of the (necessary) assumptions about the data (e.g., fluorescence microscopy) and desired information (e.g., off-target vs on-target binding) are quite specific to CoSMoS experiments and therefore limit the direct applicability of Tapqir for the analysis of other single-molecule microscopy techniques. With that in mind, the direct Bayesian inference-based analysis of image data, as opposed to integrated time series, as demonstrated here is very powerful, and may encourage and inspire related methods to be developed.

Our approach is a powerful way to analyze CoSMoS data in part *because* it is specific to CoSMoS – it is premised on a physics-based model that incorporates known features of CoSMoS experiments. We agree that the general approach could be adapted to other image analysis applications.

Reviewer #2 (Recommendations for the authors):– Some of the language in the introduction is a little imprecise (e.g., "binders", "green RNA", "blue DNA spot", "integrating binder fluorescence", "real fluorescent spots"), and could be more explicit to improve clarity.

We have edited the Introduction to improve clarity.

– Line 63: The concentration barrier could be described more in depth for the eLife readership.

The background noise problem discussed at line 63 (we don’t call it a barrier) has multiple causes, not all of which are well understood, which are only peripherally relevant to our manuscript. We think more discussion of the subject would be distracting and we instead cite the Peng and van Oijen papers which discuss the subject in more detail.

– Line 74-76: Additional description of these effects, perhaps mathematically or through other citations, would help the readers understand the fundamental differences between analyzing image data and intensity data.

The advantages of analyzing 2-D images rather than integrated intensity time series have been already discussed in the published literature by Friedman and Gelles, 2015 and Smith et al., 2019, both of which are cited (line 78).

– Line 82-83: Describing how and/or the magnitude of the failure that not accounting for spot confidence creates would be useful for the reader to understand the requirement for Tapqir

We believe that the description in the cited lines explains why accounting for spot confidence is important in principle. No previous analysis method has measured spot confidence, so there is no existing data prior to this manuscript that measures the failure magnitude.

– Line 84-86: The method described doesn't get a name, but the software does get a name. I think giving the method a descriptive name (e.g., an acronym) would help clarify the discussion and distinction between the approach of probabilistic modeling of the data and using pyro and the chosen priors etc. to do so.

We have edited the manuscript to adopt this recommendation. We now call the mathematical model “the *cosmos* model” and use “Tapqir” to refer to the software.

– More referencing of Bayesian image analysis methods for microscopy data, at least in the introduction, (e.g., Bayesian Analysis of Blinking and Bleaching (B3), maybe some super-resolution methods, etc.) would help create the appropriate context for Tapqir.

We added information mentioning prior use of Bayesian methods in single-molecule microscopy to the Discussion (line 426).

– A discussion of the benefits of variational as opposed to exact inference is missing and would be useful for the reader.

We added a brief discussion of this point (line 632).

– Line ~139: It is unclear if the image models or PSFs are integrated over pixel boundaries (i.e., as in Smith et al., "Fast, single-molecule localization that achieves theoretically minimum uncertainty" DOI: 10.1038/nmeth.1449). If not, what effect does this have on the modeling?

No, to reduce the amount of computation required, we use an approximation based on the intensity value at the center of the pixel. Use of the approximation is justified by the performance of our algorithm measured against simulated data (Figure 4, Figure 5, Figure 3 —figure supplement 2, and Figure 3, figure supplement 3).

– Line 155-161: A discussion of EMCCD versus sCMOS noise differences, or even which one is more applicable to Tapqir, would be helpful here.

This point is now addressed in the manuscript; please see our response to Essential Revisions point 1.

– Line 181: It is unclear what the "hierarchichal Bayesian analysis" refers to. I could not find an explanation in the Methods.

To clarify this point, we added (line 188) citation to Chapter 5 in Gelman et al., (2013), specifically indicating that it covers the topic of hierarchical models.

– Figure 3: What is the criteria for not having {x,y,h} included on the plot (e.g., at t=101)? I could not find it. Maybe p(m=0)>.5?

Yes, that is correct. We have edited the figure caption to make this clearer.

– Figure 3: This figure should also include w along with x,y, and h. Is it relatively constant? Does it vary quite a bit?

We have added *w* to Figure 3. It is relatively constant.

– Figure 3-supplement 1 A: It was unclear to me why at frame 103, the spot was detected as spot 2 and not spot 1 with equal probability. Isn't there a degeneracy between the two spots? Is this broken by \theta? Regardless of the answer, perhaps a more in-depth discussion of this point would be useful.

We explain in the Figure 3 caption: “Spot numbers 1 (blue) and 2 (orange) are assigned arbitrarily and may change from fame to frame.” We have added language to the Figure 3 —figure supplement 1 caption to re-emphasize that the spot 1 and 2 labels are arbitrary.

– Figure 3-supplement 3 D: How does this plot compare to the theoretical minimum uncertainty of localizing a single molecule (i.e., the Cramer-Rao lower bound) at these photon fluxes? Shouldn't it bottom out at some point?

σ^xy^ is not the precision in the position of the binder spot, it instead reflects the width of the distribution of binder spot position with respect to the target position across all AOIs and all frames. Nevertheless, the reviewer is correct that it should be greater than or equal to the localization uncertainty for an individual spot. However, even for the point in Figure 3 —figure supplement 3D with the smallest number of photons, the lower bound on localization certainty is 0.11 pixels. This value is much smaller than the range of values shown in the figure, so the “bottoming out” is not seen in the plot.

– Line 214: "… rich enough to accurately capture …" is a very nice way to convey the utility of the model. I think you should use it more often.

Thank you.

– Figure 5C-E: the rate constants are systematically overestimated -- even at the slowest rates. Why?

We explain this at lines 326-328: “At higher nonspecific binding rates, rare interruptions caused by false-positive and false-negative spot detection shorten Δ*t*_on_ and Δ*t*_off_ distributions, leading to moderate systematic overestimation of the association and dissociation rate constants.”

Might these rates constants actually transition probabilities? I did not see a k=-ln(1-P)/dt equation in the Methods section.

We have edited the manuscript to clarify that (lines 650-651) “…*k*_on_ and *k_off_* are transition probabilities that numerically approximate the pseudo-first-order binding and first-order dissociation rate constants in units of s^-1^, respectively, assuming 1 s/frame.”

– Line 353: Generally, Tapqir is only quantitatively compared to spot-picker, when there is also the Carlas et al., method that could be used for comparison.

Please see our response to Essential Revision 5.

– Figure 6B and supplements: Why were no off target controls ever analyzed to be included in the plot B as the yellow curve in C. If nothing else, it would be very useful to show the Tapqir is very accurate.

There is no equivalent to the yellow curve in the Tapqir analysis. Tapqir jointly analyzes the on- and off- target data sets, rather than analyzing them separately as in the approach used with spot-picker. The results in Figure 4I show that there are essentially no false positives seen in analyses of simulated datasets that are constructed to have no target-specific binding.

– Table 1: The computation times are reasonable for a high quality analysis, but are done on a very fast desktop computer (Threadripper with a 2080). It would be useful to show the performance on a less powerful computer as well (e.g., a low-powered laptop) for a least one dataset or perhaps a partial dataset. That way, potential users can judge whether they need to seek out better computational resources before trying Tapqir.

We have now added to Table 1 data showing that computation time on the Google Colab Pro cloud service is actually faster than that on our local GPU system. Colab Pro is readily accessible (available to anyone at $50/month) and user friendly. We have added to the user manual a tutorial that shows how to run a sample data set using Tapqir on Colab.

Reviewer #3 (Public Review):In this manuscript, the authors seek to improve the reproducibility and eliminate sources of bias in the analysis of single molecule colocalization fluorescence data. These types of data (i.e., CoSMoS data) have been obtained from a number of diverse biological systems and represent unique challenges for data analysis in comparison with smFRET. A key source of bias is what constitutes a binding event and if those events are colocalized or not with a surface-tethered molecule of interest. To solve these issues, the authors propose a Bayesian-based method in which each image is analyzed individually and locally around areas of interest (AOIs) identified from the surface tethered molecules. A strength of the research is that the approach eliminates many sources of bias (i.e., thresholding) in analysis, models realistic image features (noise), can be automated and carried out by novice users "hands-free", and returns a probability score for each event. The performance of the method is superb under a number of conditions and with varying levels of signal-to-noise. The analysis on a GPU is fairly quick-overnight-in comparison with by-hand analysis of the traces which can take days or longer. Tapqir has the potential to be the go-to software package for analysis of single molecule colocalization data.The weaknesses of this work involve concerns about the approach and its usefulness to the single-molecule community at large as wells as a lack of information about how users implement and use the Tapqir software. For the first item, there are a number of common scenarios encountered in colocalization analysis that may exclude use of Tapqir including use of CMOS rather than EM-CCD cameras, significant numbers of tethered molecules on the surface that are dark/non-fluorescent, a high density/overlapping of AOIs, and cases where event intensity information is critical (i.e., FRET detection or sequential binding and simultaneous occupancy of multiple fluorescent molecules at the same AOI). In its current form, the use of Tapqir may be limited to only certain scenarios with data acquired by certain types of instruments.

Concerns about application to CMOS, dark target molecules, and overlapping AOIs are addressed in the revised manuscript and in the responses to Essential Revisions 1, 2, and 4. The question of applying the approach to methods (e.g., smFRET) that require extraction of both colocalization and intensity data is discussed below; please see our response to Reviewer #3 (Recommendations for the authors) point 1.

Second, for adoption by non-expert users information is missing in the main text about practical aspects of using the Tapqir software including a description of inputs/outputs, the GUI (I believe Taqpir runs at the command line but the output is in a GUI), and if Tapqir integrates the kinetic modeling or not.

This information is given in the online Tapqir documentation, which is now cited in the manuscript. The kinetic analysis (as in Figure 6) is a simple Python script that is run after Tapqir; the instructions for using it are included in the documentation. Tapqir runs can be initiated using either a CLI or GUI. Output can be viewed in Tensorboard, in a Tapqir GUI, and/or passed to a Jupyter notebook or Python script for further analysis, plotting, etc.

Given that a competing approach has already been published by the Grunwald lab, it would be useful to compare these methods directly in both their accuracy, usefulness of the outputs, and calculation times.

Please see our response to Essential Revision 5.

Along these lines, the utility of calculating event probability statistics (Figure 6A) is not well fleshed-out. This is a key distinguishing feature between Tapqir and methods previously published by Grunwald et al. In the case of Tapqir, the probability outputs are not used to their fullest in the determination of kinetic parameters. Rather a subjective probability threshold is chosen for what events to include. This may introduce bias and degrade the objective Tapqir pipeline used to identify these same events.

This comment reflects a misunderstanding. No probability threshold is used in the kinetic analyses (Figures 5 and 6). Instead, we make full use of the *p*(specific) probability output using the posterior sampling strategy that is illustrated in Figure 5B and is described in the Results and in Materials and methods. We have edited the Results section to further emphasize this point.

Finally, the manuscript could be improved by clearly distinguishing between the fundamental approach of Bayesian image analysis from the Tapqir software that would be used to carry this out.

We have edited the manuscript to adopt this recommendation. We now call the mathematical model “the *cosmos* model” and use “Tapqir” to refer to the software.

A section devoted to describing the Tapqir interface and the inputs/outputs would be valuable. In the manuscript's current form, the lack of information on the interface along with the potential requirement for a GPU and need for the use of a relatively new programming language (Pyro) may hamper adoption and interest in colocalization methods by general audiences.

As described above, description of the interface and inputs/outputs is given in the online Tapqir documentation, which is now cited in the manuscript. As described above, users do not need to own a GPU, they can run the program on a readily available cloud computing service. Users do not need any knowledge of Pyro to use Tapqir; Pyro is merely used internally in the coding of Tapqir.

Reviewer #3 (Recommendations for the authors):1. It is unclear if intensity information is used by Tapqir or if it can be used. This can be useful for including more priors about the experiment (i.e., "real" events would be above a certain threshold due to FRET or presence of multiple fluorophores) or for using Tapqir to analyze experiments in which multiple fluorophore-labeled molecules bind simultaneously and sequentially to the same AOI. As presented, it would seem that Tapqir is "blind" to these types of multiple binding events.

That is correct, at least to the extent that the *cosmos* model we describe in the manuscript does not incorporate phenomena where the spot intensity at a single target changes, such as when there is FRET or multiple binders. As we point out in the final paragraph of the Discussion, more elaborate versions of the *cosmos* model that incorporate these phenomena could be developed. This would entail implementation, optimization, and validation with simulations and real data of the new model, which is beyond the scope of the present manuscript.

2. A concern for adoption of Tapqir and appreciation of this work by general audiences involves the presentation of the method and software. I think that these should be disentangled from one another and that Tapqir should only be used to refer to the software used to carry out this approach. The manuscript, and the colocalization field, may be better served if a section were included that explicitly describes how to use Tapqir to implement this analysis including the necessary inputs, hardware (how much time would this take if a GPU isn't used?), and outputs/GUI. Ultimately, Tapqir needs to be user-friendly to be adopted and the requirement for a GPU and the Pyro programming language may be significant barriers. A potential model for the authors to consider is the eLife paper describing cisTEM software (https://elifesciences.org/articles/35383) that efficiently describes both the process, benchmarking, and software/user experience.

We have addressed these points above in responses to multiple parts of the Reviewer #3 (Public Review). It is perhaps worth emphasizing that, as detailed in the first section of Results (“Data Analysis Pipeline”), we do not describe in our manuscript a full data analysis pipeline like that in the cisTEM paper but only one module in that pipeline.

3. With respect to inputs, the need for use of imscroll to identify AOIs, drift correct, and carry out mapping should be clarified. Is imscroll output essential for Tapqir input?

As discussed in the “Data Analysis Pipeline” section of Results, there are variety of tools in current use to do the pre-processing steps of CoSMoS data analysis. Unfortunately, there is no standard file format for the output of the preprocessing results, and Tapqir currently reads only the format produced by imscroll. We hope to work with users of other preprocessing tools to develop and test code that will read some of the other formats in use.